# EGUSPHERE-2024-327 | Research article

Received: 02 Feb 2024 – Discussion started: 07 Feb 2024 – Revised: 16 August 2024

Marine cloud base height retrieval from MODIS cloud properties using machine learning

Julien Lenhardt, Johannes Quaas, and Dino Sejdinovic

https://egusphere.copernicus.org/preprints/2024/egusphere-2024-327/

# Marine cloud base height retrieval from MODIS cloud properties using machine learning

Julien LENHARDT [1], Johannes QUAAS [1,2], Dino SEJDINOVIC [3]

[1]Leipzig Institute for Meteorology, Leipzig University, Leipzig, Germany
[2]ScaDS.AI - Center for Scalable Data Analytics and Artificial Intelligence, Leipzig University, Humboldtstraße 25, 04105 Leipzig, Germany
[3]School of Computer and Mathematical Sciences & Australian Institute for Machine Learning, University of Adelaide, Adelaide, Australia
*Correspondence to:* Julien LENHARDT (julien.lenhardt@uni-leipzig.de)

**Abstract**

Clouds are a crucial regulator in the Earth's energy budget through their radiative properties, both at the top-of-the-atmosphere and at the surface, hence determining key factors like their vertical extent is of essential interest. While the cloud top height is commonly retrieved by satellites, the cloud base height is difficult to estimate from satellite remote sensing data. Here we present a novel method called ORABase (Ordinal Regression Autoencoding of cloud Base) leveraging spatially resolved cloud properties from the MODIS instrument to retrieve the cloud base height over marine areas. A machine learning model is built with two components to facilitate the cloud base height retrieval: the first component is an autoencoder designed to learn a representation of the data cubes of cloud properties and reduce their dimensionality. The second component is developed for predicting the cloud base using ground-based ceilometer observations from the lower dimensional encodings generated by the aforementioned autoencoder. The method is then evaluated based on a collection of co-located surface ceilometer observations and retrievals from the CALIOP satellite lidar. The statistical model performs similarly on both datasets, and notably on the test set of ceilometer cloud bases where it exhibits accurate predictions in particular for lower cloud bases and a narrow distribution of the absolute error, namely 379 m and 328 m for the mean absolute error and the standard deviation of the absolute error respectively. Furthermore, cloud base height predictions are generated for an entire year over ocean, and global mean aggregates are also presented, providing insights about global cloud base height distribution and offering a valuable dataset for extensive studies requiring global cloud base height retrievals. The global cloud base height dataset and the presented models constituting ORABase are available from Zenodo (Lenhardt et al., 2024).

## 1 Introduction

Clouds play a key role in the Earth's energy budget through their interactions with incoming shortwave and outgoing longwave radiation fluxes. It is thus critical to adequately quantify cloud radiative properties and their changes under global climate change. However, cloud radiative properties remain a large uncertainty in estimating anthropogenic climate change and possible impacts in the future (Boucher et al., 2013; Forster et al. 2021). Radiative properties of clouds are related to numerous quantities that can be used to characterise them. For instance, the cloud base height (CBH) is a crucial radiative property through its impact on the surface longwave radiation. Furthermore, the cloud geometrical thickness (CGT), defined as the difference between the cloud top height (CTH) and the CBH, links to the adiabatic cloud water content allowing the quantification of the cloud's subadiabaticity. Additionally, deriving the CBH is of practical use for pilots, providing crucial information during flights.

However, while the CTH can be rather easily obtained through passive satellite observations, the CBH retrieval remains problematic due to the fact that it is only indirectly accessible to satellites, and due to retrieval errors related to satellite remote sensing such as instrument shortcomings or noisy measurements. Since the difference between the CTH and the CBH quantifies the vertical extent of a cloud, one way to retrieve the CBH from passive satellites is by making heavy assumptions on the vertical distribution of the cloud water path inside the cloud profile. It is thus a challenging retrieval with passive satellites data that provide information about the cloud top (e.g. cloud top temperature (CTT), pressure (CTP) or height (CTH)) or about the entire column (e.g. cloud optical thickness (COT)) assuming the cloud's adiabaticity. For example, Noh et al. (2017) rely on a semiempirical approach to link the CGT to the CTH and the cloud water path (CWP, includes both ice and liquid water paths). In a different approach, Böhm et al. (2019) retrieve the CBH from triangulation of a multi-angle spectroradiometer. However, in this case, assumptions were required on the distribution of convective clouds. On the other hand, active satellite remote sensing retrieves information with vertical resolution which greatly helps resolving the clouds vertical distribution. However, active satellite measurements can display attenuated signals close to the surface (Tanelli et al., 2008; Marchand et al., 2008) particularly in the presence of thick clouds or precipitation, rendering the retrieval of the CBH difficult even for radar and lidar. Among others, Mülmenstädt et al. (2018) and Lu et al. (2021) present methods focusing on low clouds which use the CBH from active satellite retrievals of neighbouring thin clouds as representative of the surrounding cloud field. Active remote sensing additionally suffers from the sparse sampling that is confined to a narrow swath below the satellite. Finally, Goren et al. (2018) combine information from both passive and active satellite remote sensing and rely upon an adiabatic cloud model to derive the CBH. The retrieval of the CBH using satellite remote sensing data relies on a number of simplifying assumptions and is, consequently, prone to errors. Subsequently, uncertainties in the estimation of the CBH propagate into uncertainties in the overall cloud radiative effect (CRE) (Kato et al., 2011; Trenberth et al., 2009).

The method presented here called ORABase (Ordinal Regression Autoencoding of cloud Base) leverages passive satellite retrievals of cloud properties in combination with marine surface observations to derive the CBH of a cloud scene using a machine learning (ML) model. The CBH retrieval method relies on level 2 satellite data, namely three different cloud properties which are CTH, COT and CWP. A convolutional neural network (CNN, LeCun et al., 1989; LeCun et al., 1995) model following the autoencoder (AE; Kramer, 1991; Hinton et al., 2006) framework is trained in a self supervised way to reconstruct the previously mentioned cloud properties. This type of artificial neural network has been widely used in computer vision (Krizhevsky et al., 2012; LeCun et al., 2010) but also more recently in various applications in climate science (Reichstein et al., 2019; Watson-Parris et al., 2022). Thereafter, an ordinal regression (OR; Winship et al., 1984) model is fitted to predict the CBH corresponding to the cloud properties, learning from ground-based marine CBH retrievals. These different steps constituting the method are summarised in Figure 1 and detailed in section 2. The objective of the developed method is primarily to produce CBH retrievals with reduced uncertainty, and additionally to provide extended spatial and temporal coverage compared to surface observations. Indeed, we hypothesise that the spatial pattern of the cloud field carries information about the CBH and that the CNN can exploit the potential non-linear relationship between the CBH and the satellite observations. Furthermore, as more accurate CBH retrievals are obtained from ground-based remote sensing observations which are only available at isolated locations, we capitalise on these retrievals to develop a satellite-based retrieval algorithm capable of generalising to global distributions. We sensibly reduce the scope of the study by focusing on lower clouds, in particular as ground-based CBH observations display higher accuracy compared to satellite-based retrievals in those cases, and as it is the lowest cloud which often matters most for e.g. the surface radiation budget. We also restrict the retrievals to marine regions to remove the impact of orography on surface observations especially for these same low level clouds.

Section 2 firstly introduces the datasets and the co-location between ground-based observations and satellite retrievals. Secondly, the ML method constituting ORABase is described. In section 3 we evaluate our predictions against other methods including Noh et al. (2017) and other products from active satellite measurements like the 2B-CLDCLASS-LIDAR product (Sassen et al.,

2008). Section 4 presents the global dataset of the CBH which is derived from the ML approach. We discuss the benefits and remaining challenges of our method in section 5. Further details about the spatial distribution of the observations and the ML method are included in the appendices A-E. Additional links to available data outputs and codes are listed in the corresponding sections.

## 2 Data and methods

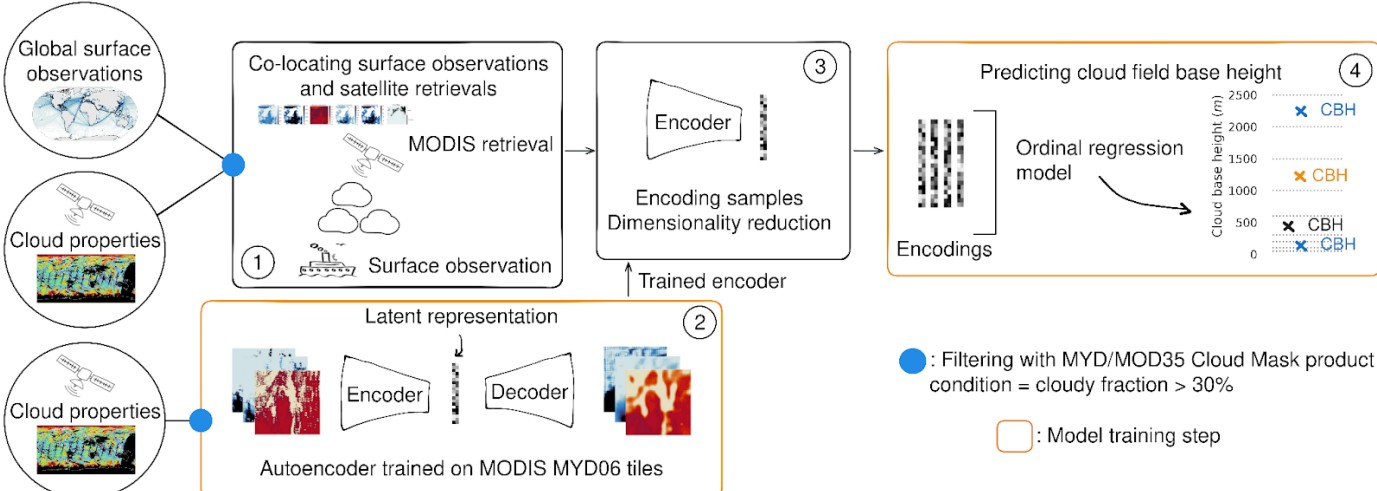

**Figure 1: Schematic of the cloud base height retrieval method. 1) Co-location of surface-based cloud base height observations and satellite retrievals. 2) Autoencoder training on satellite cloud properties. 3) Encoding of co-located samples using the trained encoder. 4) Prediction of the cloud field base height.**

### 2.1 Surface observations

The CBH labels used in this study are part of a global marine meteorological observation dataset maintained by the UK Met Office (Met Office, 2006; Table 1), which provides observational data ongoing from 1854. The observations are conducted from measuring stations that were located on ships, buoys or platforms. As a consequence, this study largely relies on observational data representing the areas along the corresponding ship routes (Fig. 2a). Despite their coarse resolution, the reported cloud base observations provide valuable information about clouds in remote marine areas. The distribution of CBH observations and corresponding bins are shown in Figure 2.

At the beginning of meteorological and weather reports, surface-based cloud observations were retrieved manually or visually by human observers, but they have been gradually replaced by automated systems. In the surface observation dataset used in the study, the CBH is derived using a ceilometer, an instrument based on a laser pointing upright and measuring the backscatter from the cloud base, and is then reported following the current standards from the World Meteorological Organisation (WMO; WMO, 2019). The CBH observations are sorted into bins of increasing width (from 50 m to 500 m bin width) corresponding to the altitude (Fig. 2b) as the data transfer through radio limits the amount of transferable information and precision close to the surface is of importance notably for aircrafts. Since the actual measured CBH values are not available in the dataset, it is impossible to directly quantify a possible bias stemming from this binning process. In general here, we can suspect that the available CBH retrievals represent an accurate or underestimated assessment of the effective CBH, as for example a ceilometer measuring a CBH of 2490 m will be reported in the 2000 m bin in the available dataset. Using for example the central value of each bin could be another way to compute averages to potentially alleviate this unknown bias but it is not presented here. However, the method presented in the following sections predicts the CBH in corresponding bins, so it is left to the user to use these as they see fit for further analysis.

| Data product | Description | Variables | Resolution | Usage |
|---|---|---|---|---|
| Global marine meteorological observations (Met Office, 2006) | Surface observations | Cloud base height (m) | Latitude/longitude coordinates 0.1° Hourly/daily observations | Labels |
| MODIS Atmosphere L2 Cloud Product (MYD06) (Platnick et al., 2017) | Cloud-top properties, cloud optical and microphysical properties | Cloud top height, CTH (m) Cloud optical thickness, COT (a.u.) Cloud water path, CWP (g.m$^{-2}$) | 1 km pixel resolution Daily overpass | Input features |
| MODIS Atmosphere L2 Cloud Mask Product (MYD35) (Ackerman et al., 2017) | Cloud pixel flag | Cloud mask | 1 km pixel resolution Daily overpass | Used for cloud scene filtering |

**Table 1 : Dataset description. The surface observations are provided by a worldwide station network available from the UK MetOffice (Met Office, 2006; cf. section 2.1). The MODIS data are derived from the collection 6.1 of the datasets (Platnick et al., 2017; Ackerman et al., 2017; cf. section 2.2).**

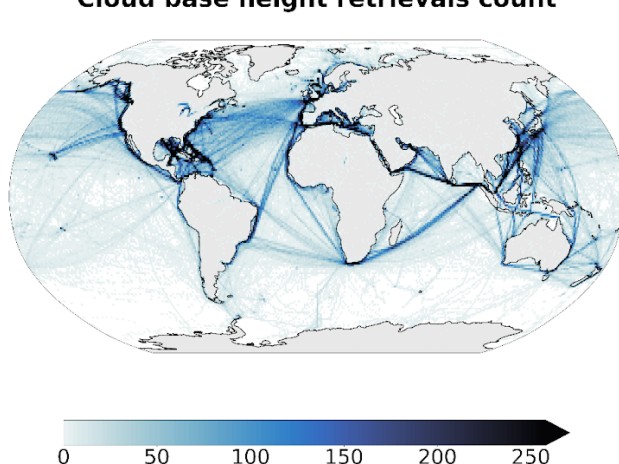

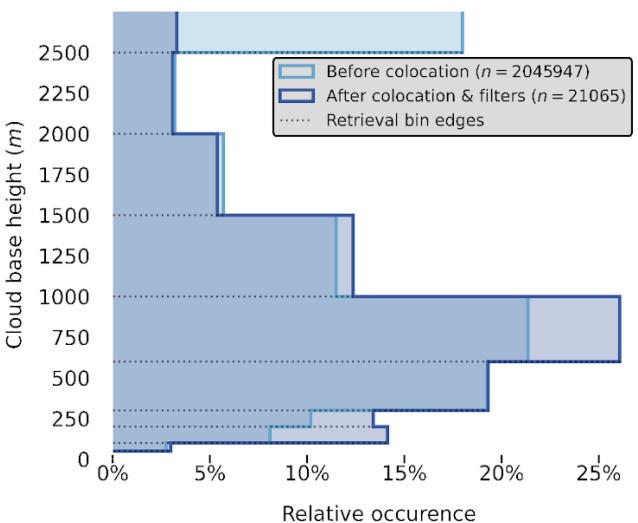

**Figure 2: (a) Spatial distribution of cloud base retrievals count (1 ° grid) and (b) distribution of the retrieved cloud base height before and after the co-location and filtering process, for observations from the years 2008 and 2016.**

## 2.2 Satellite data

In this study we use products from the MODerate Resolution Imaging Spectroradiometer (MODIS, Platnick et al., 2017) from the AQUA satellite as input data that is later combined with the CBH labels derived from the surface-based observations to train the prediction model. We choose MODIS satellite retrievals as they provide a large amount of data with kilometre-scale resolution and daily overpasses, the spatial coverage of one granule representing an area of 2330 km x 2000 km. We make use of the CUMULO dataset (Zantedeschi et al., 2019) since it provides already preprocessed satellite data from the A-train with daily full coverage of the Earth for the years 2008 and 2016. In particular out of the available variables we use two aligned products (cf. Table 1), namely the MODIS06 level 2 cloud product (hereafter MYD06; Platnick et al., 2017) which provides relevant

cloud properties and the MODIS35 level 2 cloud flag mask (hereafter MYD35; Ackerman et al., 2017) which allows us to filter scenes and screen for clouds.

The MYD06 product contains various cloud top properties (temperature, pressure, height) and cloud optical and microphysical properties (optical thickness, effective radius, water path). Level 2 data are derived from calibrated radiances through various algorithms and physical relations detailed in Platnick et al. (2017). The cloud top quantities are derived from radiance data of several channels. Wavelengths in the $CO_2$ absorption range are particularly used to identify the cloud top pressure (CTP) and thus the CTH of high clouds because of the opacity of $CO_2$. For thicker or low boundary layer clouds, since the CO2 slicing technique fails,the CTH is retrieved using the 11 µm brightness temperature band and combined with simulated brightness temperatures based on vertical profiles from GDAS using surface temperature together with monthly averaged lapse rate data (Baum et al., 2012). The use of monthly averaged lapse rate data separately for different regions greatly helped reduce the bias in retrieved CTHs for low clouds in the Collection 6 of MYD06 from Collection 5, but some spatial and regional biases remain. These biases directly impact the spatial and temporal distribution of CTH in the data and thus what the model could learn from. The cloud optical thickness (COT) and cloud effective radius (CER) are simultaneously derived from multispectral reflectances, cloud masks, CTP data and surface type characteristics. The cloud water path (CWP) is additionally retrieved as part of the cloud optical properties algorithm described in Platnick et al. (2017). The retrieval of these cloud properties additionally requires inputs such as temperature, water vapour and ozone profiles from NCEP GDAS (Platnick et al., 2003; Baum et al., 2012) which can lead to potential uncertainties in particular in remote marine regions where only sparse observations are available for assimilation.

In general, the MYD06 level 2 product offers the advantage that the statistical model can be built relying on cloud properties and it can thus allow the study of relationships between the CBH and other cloud properties. Calibrated radiances, one step ahead in the data processing pipeline, would also provide insightful information but would require inputs of larger dimensionality since key information about clouds would be scarcer. Furthermore, using MYD06 level 2 data allows us to compare our method to others which in most cases use cloud properties to retrieve the CBH. From the entirety of available MYD06 retrievals, we select three cloud properties in particular, namely the CTH, COT, and CWP. The CTH is used as it provides key information about the CBH in the cloud field, as seen in Böhm et al. (2019). Vertically integrated cloud quantities like the COT and CWP further help the statistical model by providing key information about the cloud's vertical extent, lacking in cloud top only properties, making them commonly used for retrieving the CBH (e.g. Noh et al., 2017). The CWP as computed from COT and CER, and, in consequence, also the CBH are built on adiabatic assumptions (Grosvenor et al., 2018) and therefore cannot be used to constrain subadiabaticity as also highlighted in Mülmenstädt et al. (2018).

**2.3 Datasets co-location**

We proceed to colocate our two data sources over the two years of MODIS MYD06 data available. To obtain the cloud properties of the cloud scene corresponding to the surface retrieval of CBH, we select a square tile of 128 km x 128 km from the *closest* MODIS granule available centred around the observation location. Here *closest* means that the MODIS granule contains the (latitude, longitude) coordinate of the CBH observation and the full extent of the tile centred around, and that the satellite retrieval was made during a one hour time-window before/after the CBH observation time. The spatial and temporal thresholds used to colocate the surface observations and the satellite retrievals are chosen for several reasons. Mainly, we want the satellite cloud properties to be representative of the cloud scene for which the CBH observation was made. Additionally, we want to recover a satisfying number of samples during the colocation process. Further arguments regarding the sensitivity of the retrieval method to the tile size are described in the following method section 2.5.

The extracted tile corresponding to the surface observation is then filtered. A first filter is applied to missing values in the different cloud properties fields to primarily avoid retrievals of poor quality. This is predominantly the case for the COT and CWP fields for which the retrieval fails more frequently, sometimes entirely. Another filtering is concordantly done using the MYD35 product for cloud cover (minimum of 30% of cloudy pixels) to ensure the cloud field was substantial enough for the colocated surface observation to be representative. Additional comments on the sensitivity of the CBH retrieval to this threshold are presented in the following section on the downstream task of CBH prediction. Throughout the quality filtering process, the missing data is one of the major factors impacting the amount of retained samples. On Figure 2, we can see that it seems to impact the clouds with higher CBHs.

The overall filtering and co-location process yields around 21 000 samples. This only represents around 1% of the initial CBH observations mainly due to the co-location process both in time and space with the MODIS overpasses. Missing values and cloud

cover filters are an additional factor in the reduced number of co-located samples. The presented co-located dataset is the basis to
build our cloud scene CBH retrieval.
**2.4 Autoencoder**
To circumvent the lack of labelled samples from which the relevant features are extracted, and to learn useful lower-dimensional
representations of the data, we add a dimensionality reduction step to our method through an unsupervised learning model. AEs
offer a wide application spectrum, ranging from preprocessing to the generation of new outputs. AEs are commonly used in
unsupervised learning settings for reducing the dimension of the input data to leverage the latent representations learned by the
model to perform clustering, classification or regression in a lower dimensional space (Baldi et al., 2012). We use classical AEs
for their simplicity and versatility, but other approaches to unsupervised latent representation learning, such as variational AEs
and its many variants, can be used in a similar fashion.In general, AEs learn to encode the given input data to produce a latent
representation of lower dimension. From the latent representation, the input data is then reconstructed. The learning process is
driven by what is called the reconstruction loss that minimises the difference between the input and the reconstructed output.
Here we use a convolutional AE architecture which is based on a CNN backbone in order to leverage the spatial structure of our
input data (Pu et al., 2016). We rely on the widely employed CNN architectures U-Net (Ronneberger et al., 2015) and VGG
(Simonyan and Zisserman, 2015), where the convolution layers are based on 3x3 filters, stacked in blocks followed by maximum
pooling layers, and mirrored for the decoder part of the model using transposed convolution layers (Zeiler et al., 2010). We adapt
the size of the input to fit our chosen tile size (128), the latent space size to 256, and use the improved Leaky Rectified Linear
Units (LeakyReLu; Maas et al., 2013) over the original ReLU (Nair and Hinton, 2010) as activation functions. The detailed
parameterization of the model is described in Appendix C. The model code was developed following implementations from the
packages *PyTorch* (Paszke et al., 2019) and *TorchVision* (TorchVision, 2016) and is included in the related Zenodo archive
(Lenhardt et al., 2024). The main goal of the AE training is then to minimise the loss function during the optimization or learning
process, and to reproduce the input data with the highest fidelity. For the loss function which in this case is only the
reconstruction error, we use the common mean-squared error (MSE), which can be written for a batch of samples as :

$$\mathcal{L}_{reconstruction} = \frac{1}{N_i} \sum_{b \in B_i} \left\| b - D_\theta(E_\theta(b)) \right\|_2^2 \qquad (1)$$

where, with the tiles used for training the AE noted as $B = \{b_n \in \mathbb{R}^{3 \times 128 \times 128}\}_{n \in [1, N]}$, $B_i$ represents a batch of samples of size
$N_i$ and $\theta$ the combined parameters of the encoder $E$ and decoder $D$ models. The MSE considered here between the inputs and
outputs of the AE is unitless, as the inputs are standardised before processing to ensure each channel is on similar scales and a
more stable model training.
However, this self supervised step requires a large amount of data that the AE can learn from. Therefore, we select one full year
of data of MODIS granules from the CUMULO dataset (from the year 2008, cf. section 2.2) and randomly sample tiles following
the same criteria as during the co-location process (cf. section 2.3). We sample a maximum of 20 tiles from a single granule and
this for only a single year of data in order to avoid possible spatial and temporal auto-correlation in the data used for training and
testing leading to a non-representative performance of the mode (Kattenborn et al., 2022). Further details on the study of the
generalisation performance of the model for new observations in space and time are given in appendix B. The overall built
dataset consists of around 500 000 samples which are then splitted for training, validation and testing based on their retrieval
date. For further testing, we additionally create a test dataset based solely on data from the year 2016 which includes tiles not
only over ocean but also over land, indicating potential generalisation skill for unseen data including orography influence. The
reconstruction error during training and validation is shown in Figure 3 along with examples of reconstructed samples. The
spatially averaged reconstruction errors per cloud property channel are displayed in Figure 4 for each of the training, validation
and testing datasets previously mentioned. The trained model reaches an MSE of 0.19 on the test set of 2008 and of 0.24 on the
global test set of 2016. The presented model is trained on tiles of size 128x128, but some arguments regarding the choice of the
tile size are made in the following section in the context of the downstream task of CBH prediction.

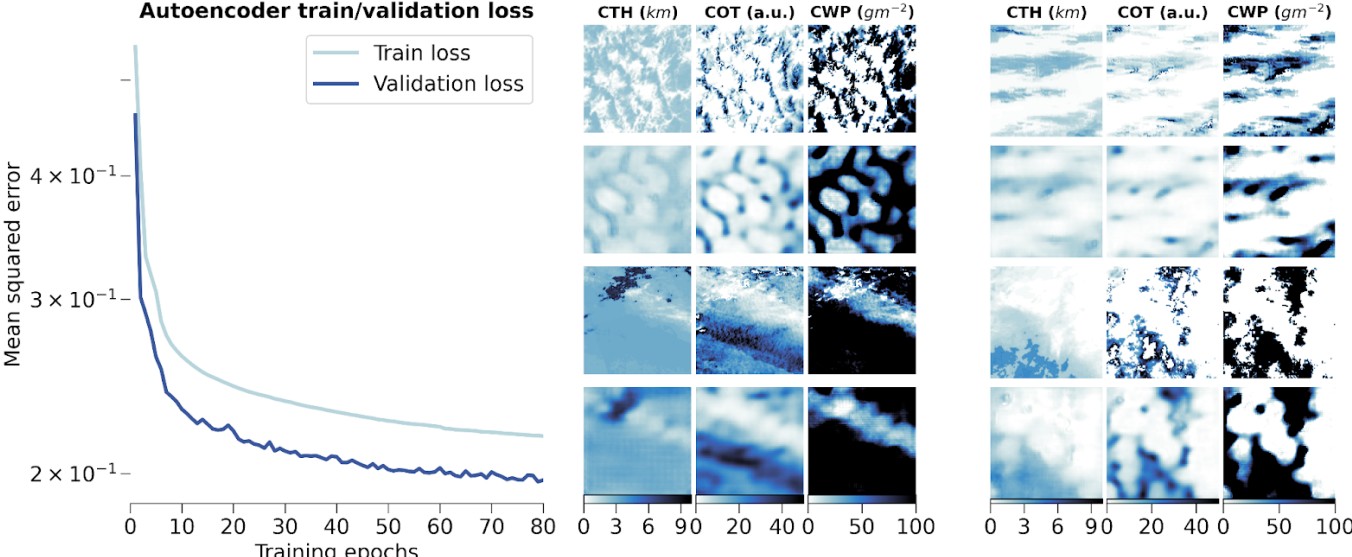

**Figure 3: (left) Training and validation losses during model optimization. (right) Examples of tiles (first and third rows) with the corresponding reconstructions (second and fourth rows) for the different cloud property channels.**

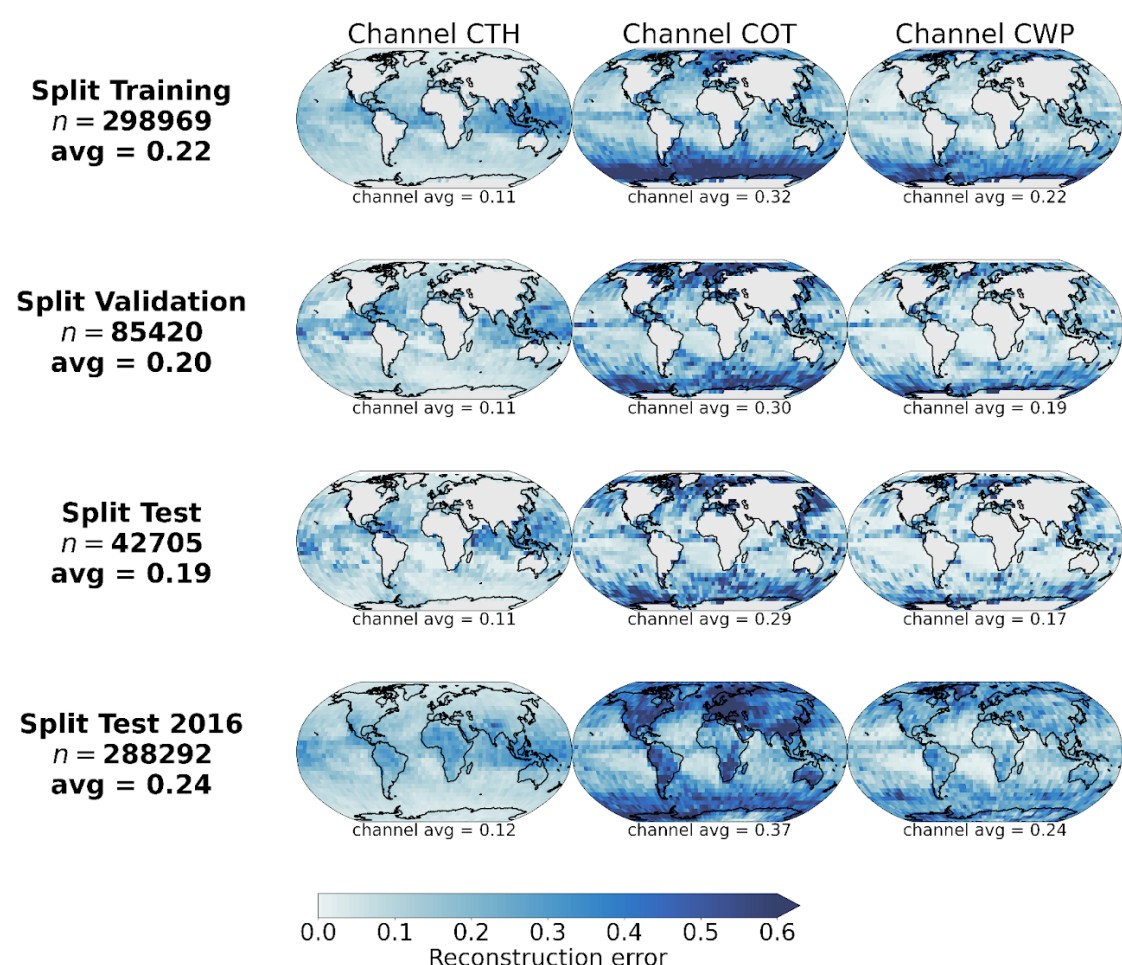

**Figure 4: Spatial distribution of channel reconstruction errors aggregated on a 5 ° grid for the 2008 training, validation, test and the 2016 test datasets.**

**2.5 Cloud base height ordinal regression**

Once the AE's optimization process is completed, the next step is to predict the corresponding CBH for the observed scene. As seen in Figure 2, the retrieved CBH observations are binned into different categories following WMO standards (WMO, 2019). This leads to a prediction problem at the intersection of regression (i.e. predicting numerical values) and classification (i.e. predicting the object class) called ordinal regression (OR). The labels from the target variable are defined by classes following a certain order, in this case the increasing CBH. A wide array of methods stems from this field with diverse applications for example in computer vision using neural networks (e.g. Niu et al., 2016; Shi et al., 2023; Lazaro and Figueiras-Vidal, 2023). Different methods exist to tackle such problem setups either via modification of the target variable, ordinal binary decomposition or threshold modelisation (Gutiérrez et al., 2016; Pedregosa et al., 2017). Threshold models were shown to be able to perform better than the ones designed for regression or multi-class classification on OR tasks (Rennie et al., 2005). We consider here two alternative frameworks in the case of threshold models which differ in how they penalise threshold violations: immediate-threshold (IT; Eq D.1) and all-threshold (AT; Eq D.2). The overall training process of the model aims at optimising a set of weights to project the input data to a one dimensional plane, subsequently dividing the constructed representation using learnable thresholds. These two implementations of threshold models are available from the *mord* Python package (based on Pedregosa, 2015) and further details on threshold OR models are added in appendix D.

To help evaluate the prediction model, we rely on a set of different metrics pertaining either to the regression aspect of the problem or to its classification/ordinal nature. First, the macro-averaged mean absolute error (MA-MAE) is used as it weights each class separately before averaging the subset MAEs, making it useful in the case of OR problems with imbalanced datasets (Baccianella et al., 2009). Using a macro-averaged metric prevents us from choosing a trivial model which might always predict the dominating class. Additionally, the macro-averaged root mean square error (MA-RMSE) is also used to investigate the skill of the prediction models. To assess the ordering of the predicted retrievals with respect to the labels, the ordinal classification index (OC; Cardoso and Sousa, 2011) and its updated version the uniform ordinal classification index (UOC; Silva et al., 2018) are computed. A version of the latter not requiring an extra hyperparameter, the area under the UOC (AUOC; Silva et al., 2018), is also reported. These different metrics are able to capture the proper ranking order of the predictions compared to the labels using the confusion matrix and also the overall accuracy of the prediction model. Nevertheless, one caveat is that these indexes developed for ordinal classification assume each class to be equally distant from another which is not the case here since the CBH retrievals are reported in bins of variable width. However, a purely ordinal classification index will drop all information on the scale of the response (1500 m misclassified as 600 m treated the same as 200 m misclassified as 50 m, since only the order matters) which might be not entirely appropriate for this problem. In an effort to address this limitation, the indexes are adapted to mimic the spacing between the different CBH bin classes by incorporating classes that are all spaced by 50 m, ranging from 50 m up to 2500 m. In this manner, the CBH class difference is more suited to the actual nature of the retrieval.

However, several aspects of the ordinal regression model need to be investigated first. To this extent, we first divide our global colocated dataset (section 2.3) in training, validation and testing datasets but while ensuring each class is relatively equally represented in each split. The following aspects and sensitivities of the model to the input data parameters are assessed using the training and validation datasets: the potential benefit of using the spatial context through the AE, the input tile size and the cloud cover threshold. Moreover, the spatial generalisation skill of the model is studied by splitting the colocated dataset between the Northern and Southern hemispheres. For each of these, the performance for the AT variant of the OR model is reported as it performs significantly better than the IT variant across experiments and evaluation metrics.

**2.5.1 Spatial context**

In order to evaluate the actual effect of the spatial context with respect to the input cloud properties, the prediction skill of the model trained based on the AE encodings is compared to a collection of three baseline methods: two trivial methods (predicting the majority bin and predicting the bin minimising the MAE across the training dataset) and an OR method relying on the flattened cloud properties of a 9x9 tile centred around the observation. Both of the trivial methods result in always predicting the CBH bin of 600 m. The third method yields a similar dimensionality as the AE encodings (3 channels x 9 x 9 = 243) and thus helps to show how the AE potentially leverages some spatial information about the cloud scene. Across all metrics, the method using the 9x9 tile input is outperformed by the OR method based on the AE encodings and even by the trivial choice of the majority bin.It is in particular noticeable with an increase of the MA-RMSE by 400 m and of the MA-MAE by 140 m compared to the OR predictions made with the AE. On the other hand, considering the predictions made with the trivial method leads to an increase of the MA-MAE of 50 m, but a decrease in MA-RMSE as most of the labels are actually concentrated around the 600 m bin. The mean bias of the trivial method is lowered closer to 0 m as it leads to a more substantial underestimation of the high

CBHs and overestimation of the low CBHs. To conclude the comparison with these two other baselines, the information spatially encoded by the AE over the whole tile size area is useful in producing CBH retrievals of better quality compared to a baseline OR model with a reduced spatial context or a trivial method predicting a singular bin.

### 2.5.2 Tile size

A prediction model is fitted to the input data using encodings produced with tailored AE models trained as detailed in the previous section but with varying square input tile sizes of 16, 64 and 128. With the subsequent prediction models, the retrievals made with a tile size of 128 showcase the lowest MA-MAE (0.8% and 2.7% decreases compared to tile sizes of 16 and 64 respectively) and MA-RMSE (around a 5% decrease compared to both other tile sizes), while no clear sensitivity arises from the OC, UOC or AUOC. Examining performance for each class separately indicates reduced errors (MAE and RMSE) for higher CBHs (above 1000 m) using the larger tile size of 128 and on par performance across tile sizes for lower CBHs. In the context of the presented CBH retrieval, the larger spatial information provided through the input tile seems to be useful for the subsequent CBH prediction task, leveraged with the help of the AE as shown previously.

### 2.5.3 Cloud cover

The colocated dataset is first filtered again with cloud cover thresholds of 10%, 20% and 30%. Each threshold respectively leads to datasets of 25 042, 23 034 and 21 065 samples which are then further splitted in training, validation and testing. On the validation set, while the decreases in MA-MAE (4.5%) and MA-RMSE (10%) with the 10% compared to the 30% cloud cover threshold are indicating a potential benefit of lowering the threshold, investigating the MAE and class-wise MAEs sheds a different picture: the benefit seems to marginally concern the higher CBH classes while hindering performances on low CBHs which overall explains the trend in RMSE notably. Considering the confusion matrices generated for each cloud cover threshold additionally shows that a lower cloud cover threshold results in a slightly increasing distribution shift of the predicted CBH classes towards higher CBHs, displaying a prediction cluster around 1000m. Overall, the benefit of additional available samples when lowering the cloud cover threshold does not seem to directly lead to convincing improved performance. The main axis of improvement here is probably lying in the widening of the colocation process to ensure broader spatial and temporal coverage of the training dataset.

### 2.5.4 Spatial generalisation

Furthermore, in a similar way as for investigating the spatial generalisation ability of the AE, we split our colocated dataset between the Northern and Southern hemispheres. This way, we ensure a minimal amount of samples in each spatial split (17 615 and 3 450 for the Northern and Southern hemispheres respectively) even though the spatial distribution patterns of the retrievals greatly differ. As a result, the lower amount of samples in the Southern hemisphere leads to some overfitting with metrics systematically worsening when testing on the Northern hemisphere. However, the Northern hemisphere training displays fair generalisation skill with equal or improved metrics when testing on the Southern hemisphere, for example an 8% decrease in MA-RMSE, 1% decrease in OC and stable MA-MAE, UOC and AUOC. The class-wise performances for the two splits reveal the overall generalisation difficulty for higher CBHs (above 600 m) when training on the Southern hemisphere, as the labels relative to these classes are mostly present in the Northern hemisphere (Figure A.3). The ability of the model to generalise from the Northern hemisphere labels reassures the overall skill of the model once trained on all the labels available.

In the following section, we present the results of the developed method alongside comparisons to previous retrieval approaches. In particular, we compare our retrieval to a method assuming an adiabatic cloud model (adapted from Goren et al. (2018), cf. appendix E for implementation) and to the method from Noh et al. (2017). The former relies on the CTH retrieved from CALIPSO's Cloud Aerosol Lidar with Orthogonal Polarization (CALIOP; Hunt et al., 2009) and CloudSat (Stephens et al., 2008), but CWP and CTT retrievals from MODIS MYD06. However, in our own comparison study we used all necessary variables, including the CTH, from MODIS MYD06. The latter method relies on piecewise linear relationships between MODIS CWP and the geometric thickness of the uppermost layer from CALIPSO/CloudSat stratified by MODIS CTH. The application of the method presented in Noh et al. (2017) is however done with CTH retrievals from the Suomi–National Polar-Orbiting Partnership (SNPP) VIIRS. The comparison to our method presented here is done by using the MODIS/CALIPSO/CloudSat-derived parameters from Noh et al. (2017), but using the MODIS derived CTH to produce the final CBH estimate. In both cases, since these methods can be applied pixel-wise when a MODIS retrieval is available, we computed the retrieved CBH values and averaged them over the cloud scene.

## 3 Results, evaluation, and comparison to previous retrieval approaches

### 3.1 Cloud base height retrieval, evaluation and comparison to previous retrievals

In this section, we present the results of the retrieval, evaluate it using the ground-based observations, and investigate how our method fares by comparing it to a method assuming an adiabatic cloud model (adapted from Goren et al. (2018), cf. appendix E for implementation) and to the method from Noh et al. (2017). The analysis is performed for the co-located scenes where ground-based observations are available. To be able to compare the relevant metrics for the different methods we proceed to a binning of the data following the WMO standard presented in section 2.1. In Table 2 we report several metrics including the MAE, the mean error (bias), the RMSE and the standard deviation of the absolute error. The latter helps us characterise the spread and uncertainty in the overall predictions with respect to the surface observations. We additionally report the adapted version of the AUOC mentioned in section 2.5. Furthermore, we do not report quantities such as the correlation coefficient or the regression line on the 2-dimensional histograms of Figure 5 and Figure 6, as the stratified and categorical aspects of the data would make reporting these not clearly informative. We refer to the overall conceived method including the AE (cf. section 2.4) and the OR prediction model in the AT variant (cf. section 2.5), listed in Table 2 as ORABase.

We first note that the OR method with an immediate-threshold setup fails at predicting adequately the cloud scene base height compared to all the other retrieval products, producing large errors (double-fold in comparison to the all-threshold setup). On the other hand, ORABase performs well with satisfying error measures and uncertainty in the predictions on par if not better than the two retrievals from Goren et al. (2018) and Noh et al. (2017). Compared to the method from Noh et al. (2017), our method succeeds in decreasing on average the error, displaying a reduction of 100 m for the MAE. The method also effectively diminishes the uncertainty in the CBH retrievals, bringing down the absolute error standard deviation 200 m lower. Our method thus provides accurate retrievals with comparatively low general uncertainty levels. Even though on average the predictions exhibit a slight positive bias, we find that the CBH values above 2000 m are systematically underestimated (Fig. 5). In consideration of the low representation of such observations in the dataset, due to data filtering and surface observations being less reliable for higher clouds, the method still struggles to properly quantify the cloud scene base height of these samples. These samples also make up for most of the measurement uncertainty in the labels considering that ceilometers face challenges for retrieving cloud signals higher up in the boundary layer. Focusing on lower cloud scene base height retrievals, the predictions demonstrate even lower errors: the MAE is lowered to 379 m while the absolute error standard deviation is narrowed down to 328 m. Achieved accuracy levels and uncertainty measures attest to a certain trustworthiness of the cloud scene base height estimates, in particular in the context of product requirements for example the ones outlined by the Joint Polar Satellite System (JPSS; Goldberg et al. (2013); 2 km accuracy threshold). However, the cloud scene base height retrieval method presented here does not aim at constituting a product on its own as it is not operational with the processing of daily new data available from the MODIS instrument, but rather at providing robust estimates of CBH for lower level clouds. Therefore, it is expected and reasonable that the accuracies and uncertainties presented here are below such thresholds. However, the available method code (Lenhardt et al., 2024) easily allows the processing of new data for users, in addition to the available dataset for the year 2016.

We performed further sensitivity studies on our retrieval method trying to improve the quality of the predictions. An attempt to balance the dataset by oversampling the higher CBH values (cloud base retrievals falling into the 2500 m bin), however, did not yield better results overall but also posed a higher risk of overfitting to these specific samples. Furthermore, any spatial information about the location of the satellite retrieval was not included as to prevent possible overfitting to the latitude and longitude coordinates of the observations present in the training data. Since the observations are sparsely distributed especially in the southern hemisphere (cf. figures from appendix A), the goal is to avoid any kind of induced spatial bias and sensitivity in the model's predictions. Accordingly we can then ensure proper generalisation skill to new spatial areas, but not only based on known retrieval distributions at similar locations. As a consequence, the choice was made to evaluate the potential generalisation skill of the prediction model by establishing a geographic distribution of the mean predicted cloud scene base height for a whole year's worth of MODIS overpasses. This is discussed in more detail in section 4. On the other hand, the temporal aspect of the model's generalisation skill was intrinsically ensured by building a test set temporally distinct from the training set, including co-located samples only from the last months of 2016.

| Method | MAE (m) | Bias (m) | RMSE (m) | Absolute error standard deviation (m) | AUOC |
|---|---|---|---|---|---|
| Goren et al. (2018) | 457 | - 262 | 689 | 515 | 0.92 |
| Noh et al. (2017) | 578 | - 35 | 860 | 638 | 0.92 |
| OR (IT) + AE | 991 | + 595 | 1296 | 836 | 0.93 |
| **ORABase** | **447** | **+ 58** | **614** | **420** | **0.89** |
| ORABase training | 456 | + 80 | 620 | 420 | 0.89 |

Table 2: Performance on the test set of different CBH retrieval methods. OR models are either built with the immediate-threshold (IT) or all-threshold (AT) variant. The method on which the rest of the study is based has been highlighted in bold and its corresponding performance on the training set is added in the last row.

**3.2 Comparison to spaceborne radar-lidar retrievals of the CBH**

The combined datasets which are part of CUMULO (Zantedeschi et al., 2019), in particular the radar and lidar retrievals, facilitate the joint evaluation of our method with both ceilometer surface observations and active satellite retrievals. Specifically we leverage the 2B-CLDCLASS-LIDAR product (Sassen et al., 2008) which is derived from the combination of CloudSat's Cloud Profiling Radar (CPR; Stephens et al., 2008) and CALIPSO's Cloud-Aerosol Lidar with Orthogonal Polarisation (CALIOP; Hunt et al., 2009). The base height of the lowest cloud layer retrieved by the instruments in each scene is considered the scene CBH and then averaged over the available pixels along the track, preserving the same spatial extent as the associated cloud properties from the MODIS instrument. For the co-located samples of the year 2008, we thus jointly retrieve the obtained CBH from the 2B-CLDCLASS-LIDAR product, only considering cases where a surface observation was in the vicinity of the satellite track (inside a disc with a ~60 km radius around the surface observation, cf. section 2.3). For the samples fulfilling these conditions, we then compare how the different retrievals fare. In Figure 6, the joint histograms for the surface observations, the 2B-CLDCLASS-LIDAR retrieval and the method's corresponding predictions are documented, representing a total of around 800 samples.

Investigating the joint histogram between the surface observations and the 2B-CLDCLASS-LIDAR retrievals (Fig. 6a) allows to identify shortcomings of the active satellite retrievals in particular close to the surface (Tanelli et al., 2008; Marchand et al., 2008). Indeed, the CBHs closer to the surface are not well captured by the 2B-CLDCLASS-LIDAR retrievals as partially expected, due to thick clouds attenuating the lidar signal, and due to ground clutter and lack of sensitivity to small droplets near cloud base for the radar signal. A similar explanation can eventually be articulated as a whole for the co-located retrievals, considering that the mean bias between the two retrievals is greater than + 600 m. Concurrently, it is fruitful to compare the 2B-CLDCLASS-LIDAR retrievals with the predictions from the developed method (Fig. 6b). As seen previously, ORABase struggles at higher CBHs, but agrees here reasonably well with the active satellite retrievals, especially for retrievals between 500 m and 1500 m. Focusing on retrievals under 1.5 km, the prediction model achieves similar performance as presented in Table 2 with a MAE of 488 m and a RMSE of 576 m, even though the subset here is much smaller.

Furthermore, we created a more extensive dataset using only 2B-CLDCLASS-LIDAR retrievals and the cloud scene predictions with the aim of obtaining a more complete view of the relationship between these two retrievals. To this extent, we collated around 160 000 samples of aligned cloud scene base height predictions and the 2B-CLDCLASS-LIDAR retrievals over the year 2016. For this dataset, the performance metrics exhibit similar values as on the previously presented subset, displaying even lower values for the MAE and the absolute error standard deviation (around a 50 m decrease for both). Similarly to the previous co-located subset, limiting the evaluation to lower cloud base retrievals yields performance metrics close to a 450 m MAE and a 270 m absolute error standard deviation, both of these being mainly impacted by agreeing retrievals in the 500 m to 1500 m range.

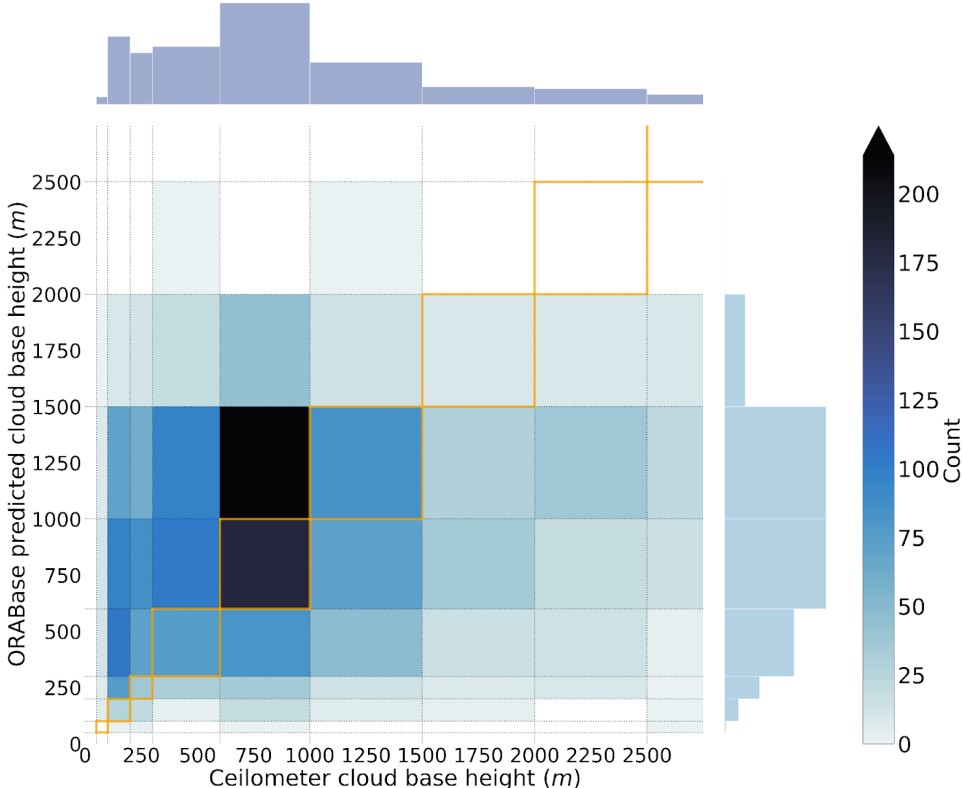

**Figure 5: Joint histogram over the test set of the surface observations and the predicted cloud scene base height from ORABAse with the ordinal regression all-threshold model. The 1:1 boxes are highlighted in orange in the figure.**

## 4 Global distribution

To further evaluate the method, we also apply the prediction model on global MODIS data for the whole year of 2016. The sampling process yields approximately 700 000 CBH retrievals for the corresponding cloud properties tiles. We then spatially aggregate the predictions to a regular grid of 5° and compute the annual mean per grid cell along the annual median absolute deviation (MAD). The MAD constitutes a useful metric to quantify the variability while removing the effects of outliers. For more robust evaluation and statistics, only ocean grid cells with more than 100 CBH retrievals over the year are displayed thus impacting mostly coastal and polar regions where filtering for ocean-only scenes or the original amount of satellite retrievals leads to a higher rate of displaying removal. The spatial distribution of the mean cloud base (Fig. 7, top) is similar to the outlined global distributions from other studies using different instruments and methods (Böhm et al., 2019; Lu et al., 2021; Mülmenstädt et al., 2018). The illustrated global quantities were established using MODIS overpasses which happen at a practically constant local time (13:30 h, early afternoon for AQUA). The MAD pattern exhibits similar characteristics (Fig. 7, bottom), even though variability slightly increases in the vicinity of land masses. These interpretations still remain valid when looking at relative deviations. Typical features are lower cloud bases towards polar regions and the mid-latitudes, and higher ones in the tropical regions. One can further observe regions like the Pacific coast of South America or the Namibian coast which display lower cloud bases concurrently with lower variability (also highlighted in Lu et al. (2021)). It is however impossible to follow up the study for nighttime retrievals, as some MODIS cloud properties are not retrieved then.

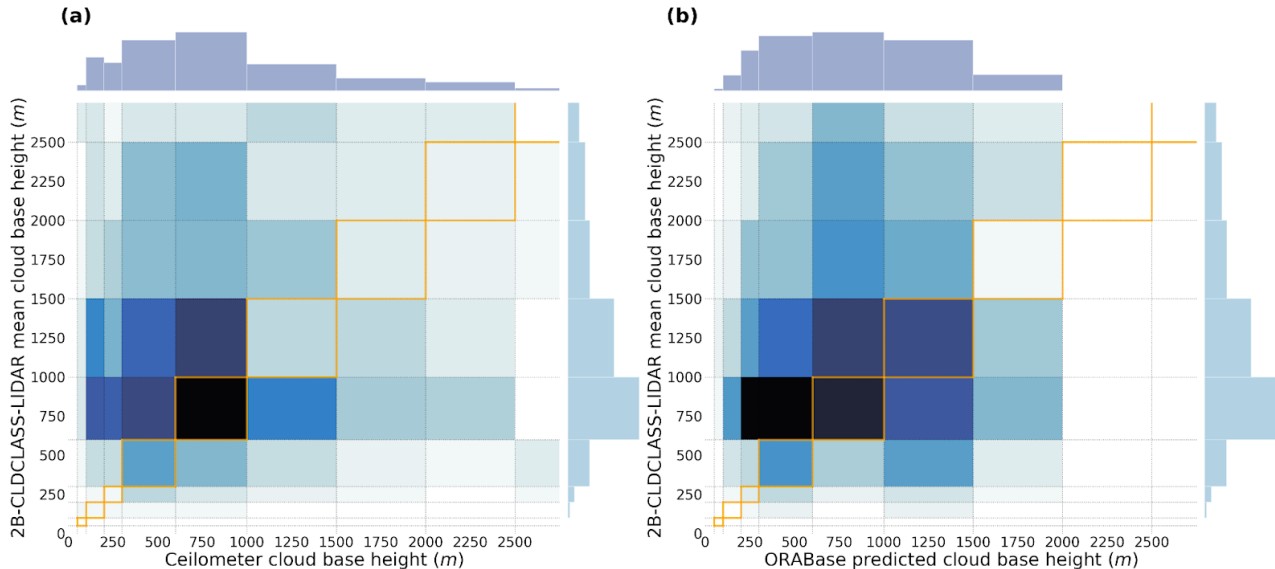

**Figure 6: Joint histogram of (a) surface observations and 2B-CLDCLASS-LIDAR retrievals, and (b) ORABase predictions and 2B-CLDCLASS-LIDAR retrievals, for the co-located cloud scenes during the year 2008. The 1:1 boxes are highlighted in the figure in orange.**

## 5 Conclusion

We have presented here a novel method named ORABase which retrieves the cloud scene base height over marine areas from MODIS cloud properties, specifically CTH, COT and CWP. This method can produce robust CBH estimates for cloud scenes in particular for lower cloud bases (MAE of 379 m and absolute error standard deviation of 328 m for up to 2 km cloud bases), based on the assumption of a homogeneous cloud base across the considered cloud field. The statistical model was built on surface observations of cloud bases with ceilometers (section 2.1), and then evaluated in comparison to other methods using passive satellite instruments (section 3.1) and active satellite retrievals (section 3.2). Analysis of the yearly averaged CBH (section 4) helped to further make sense of the predicted cloud bases and variability. The global dataset for the year 2016 is available from Zenodo (Lenhardt et al., 2024).

Using the spatially-resolved information of cloud fields of CTH, COT and CWP through the described CNN-AE results in more accurate CBH retrievals compared to the active retrievals of the 2B-CLDCLASS-LIDAR product, producing better performance metrics compared to the other products and methods considered in this study. The combination of a CNN based AE to reduce the dimensionality of the spatial patterns of cloud properties followed by a simple OR model leads to a better CBH retrieval compared to previous presented methods. The OR modelisation helps bridging the gap between regression and classification, facilitating the use of the binned cloud base observations provided by the surface observation dataset. Overall, ORABase achieves low error in the retrievals, around 400 m, and concurrently a narrow absolute error distribution, more precisely around 400 m absolute error standard deviation. Both of these performance metrics are additionally reduced when focusing on cloud bases lower than 2 km. Application to data over land areas has not been processed yet but would certainly require adding surface observations from land during the training process (e.g. Böhm et al., 2019; Lu et al., 2021; Mülmenstädt et al., 2018). Application of the presented retrieval method to other instruments could also be considered. Incorporating TERRA MODIS data would help constrain the annual mean estimates presented in Figure 7 by partially removing the potential bias of the single daily overpass arising from using only AQUA data presented in this study. The aspect enabling potential application of the retrieval method to different instruments outside of the two MODIS sensors would be the standardisation process for the input cloud properties before the use of the AE which is done based on means and standard deviations computed from AQUA-only granules. Carefully investigating the characteristics of the distribution of the cloud properties from another instrument to ensure proper scaling when using the trained AE would be then necessary. Further tests could be additionally done using coarser resolution for the input cloud properties.

Furthermore, classical semi-supervised pipelines like the one presented here, characterised by a small labelled dataset and a vast unlabelled dataset, necessitate a kind of co-location or matching process which often proves to be cumbersome and generates only a limited amount of labels. However, future avenues of research could consider directly modelling unmatched datasets, as in

e.g. Lun Chau et al. (2021) with multiresolution atmospheric data, by making use of other quantities present in the observations as mediating variables to model the link between observed and unobserved variables.

In essence, the main benefit of producing better cloud base estimates is to gain accuracy in the overall retrieval of cloud geometry, impacting in particular radiation estimates (Kato et al., 2011) like the surface downwelling longwave radiation (Mülmenstädt et al., 2018). ORAbase can thus prove to be useful by helping to produce CBH with enhanced confidence at a global scale.

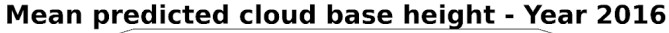

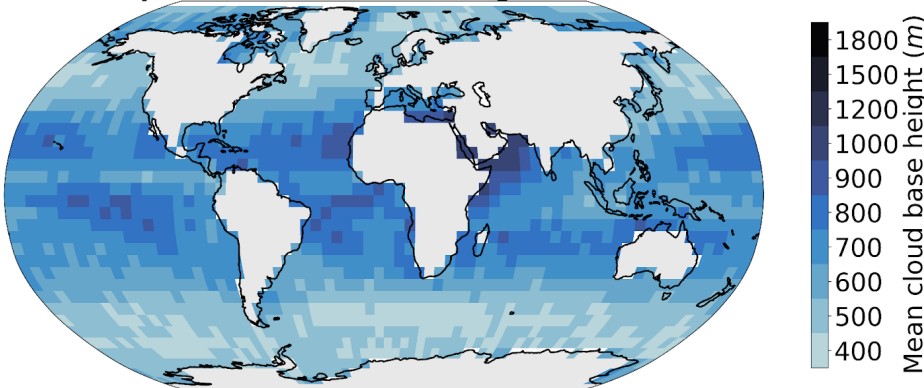

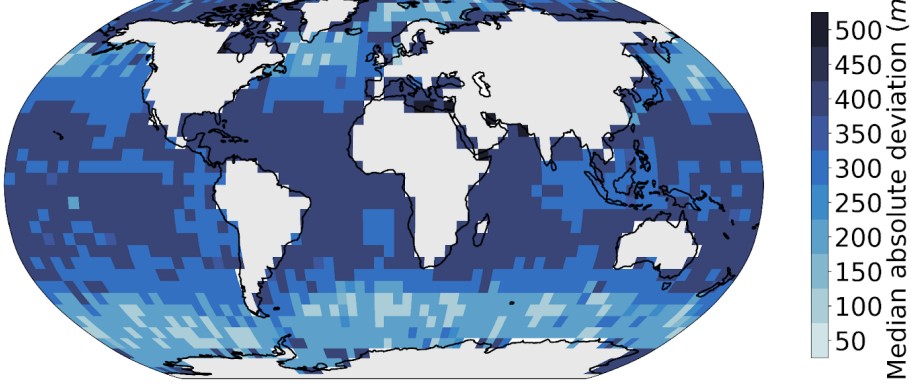

**Figure 7: Spatial distribution of (top) mean and (bottom) median absolute deviation of predicted cloud base height for the MODIS data of the year 2016 aggregated on a 5 ° grid.**

**Appendix**

**Appendix A: Cloud base height retrievals distribution**

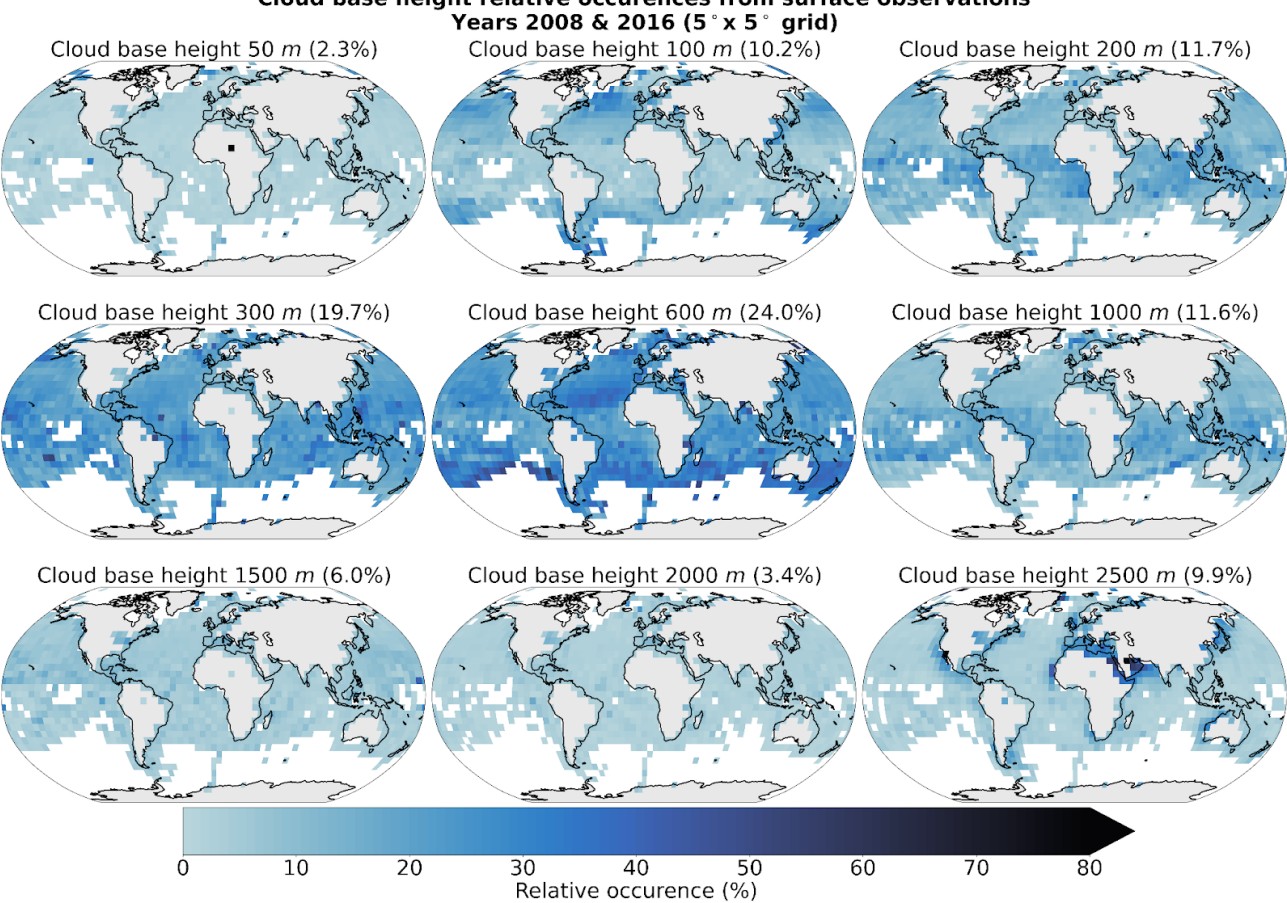

**Figure A.1: Spatial distribution of cloud base height retrievals (Met Office, 2006) for the years 2008 and 2016 on a 5 °**
**grid. Overall percentage of each label in the total observations is indicated in brackets. Only grid cells with more than 50**
**retrievals are displayed.**

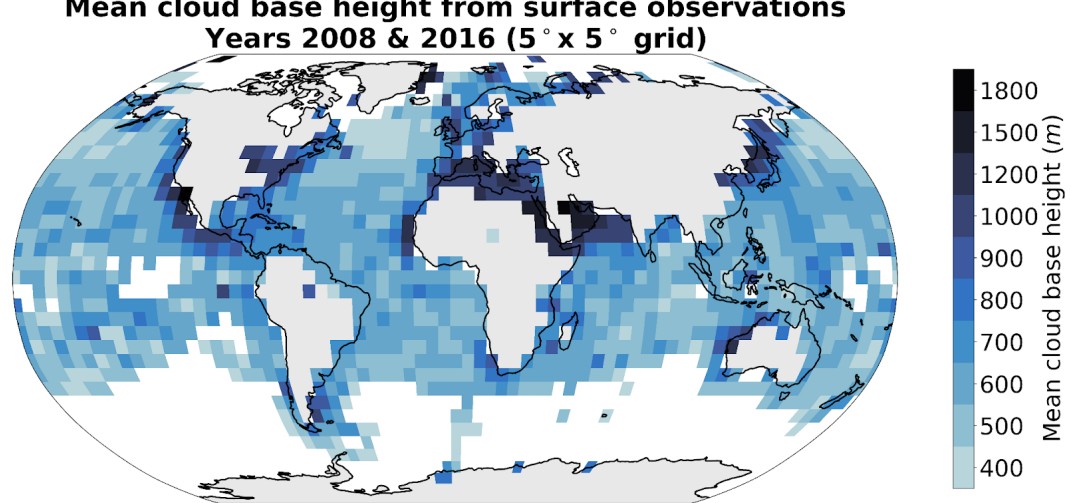

**Figure A.2: Mean cloud base height from retrievals (Met Office, 2006) for the years 2008 and 2016 on a 5 ° grid. Only**
**grid cells with more than 50 retrievals are displayed.**

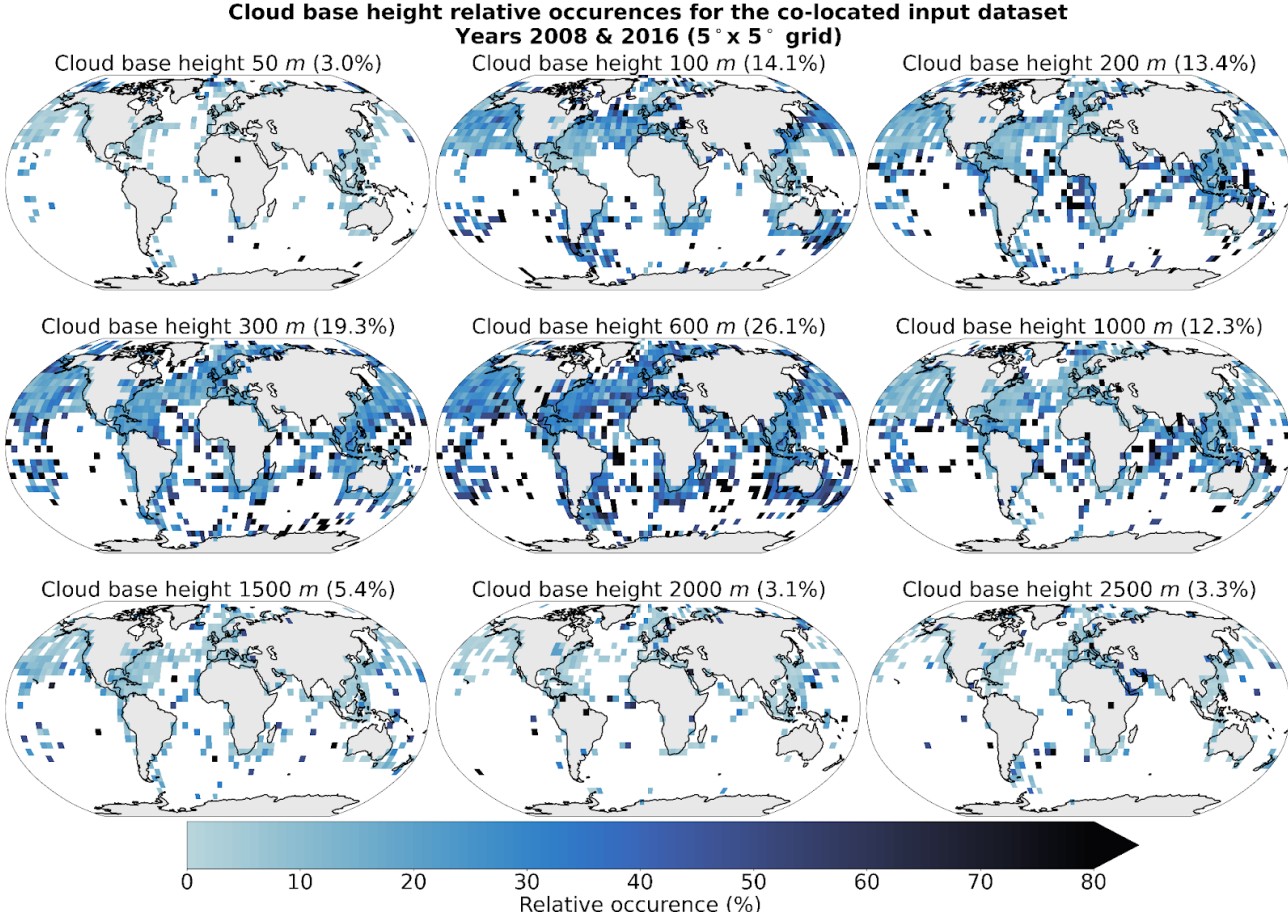

**Figure A.3: Spatial distribution of the co-located cloud base height retrievals (Met Office, 2006) and the satellite cloud properties used for training the prediction model for the years 2008 and 2016 on a 5 ° grid. Overall percentage of each label in the total dataset is indicated in brackets.**

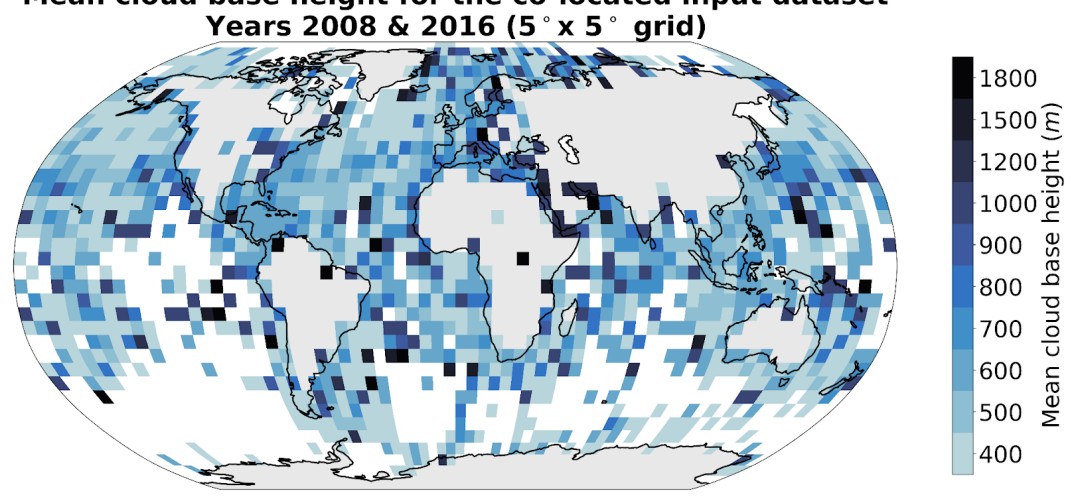

**Figure A.4: Mean cloud base height from the co-located retrievals (Met Office, 2006) and the satellite cloud properties used for training the prediction model for the years 2008 and 2016 on a 5 ° grid.**

**Appendix B: Spatio-temporal correlation study**

We create five different datasets to evaluate how the chosen AE architecture is capable of generalising to new data while trying to remove some possible autocorrelation biases which might inflate the performance scores. We also use this study to analyse how the AE model behaves when trained with our input data. We define two splits for space and time in order to build the training and testing datasets, namely the South-western (SW) quadrant and the period from March to October, respectively. The granules used to build the datasets span across the whole year of 2016. The *random* data split is the basis for the training of the model and consists of tiles sampled in the aforementioned quadrant and time period. These tiles are then split randomly between training, validation and testing datasets. This split represents the common way of splitting data when building a ML model. In contrast, we build 3 other datasets which vary through their respective spatial and time spans. The *spatial* split is built considering tiles spanning across a distinct time period, here between November and February, regardless of their spatial location. The *temporal* split is built considering tiles located anywhere but in the South-western quadrant regardless of the time at which the retrieval occurred. Finally the *spatio-temporal* split combines the previous two conditions in order to build a dataset in which the tiles come from an independent location and time as the ones used for training. Additionally, we create a global data split using data from a different year, here 2008, without any spatial restriction for the tiles. Furthermore, only a limited number of tiles was extracted from each granule while only granules from non-consecutive days were used in order to limit possible correlation between the extracted scenes.

| Data split | Time period | Spatial extent | $n$ |
|---|---|---|---|
| Random | 03-10.2016 | SW quadrant | Train: 14 691<br>Validation: 4 198<br>Test: 2 099 |
| Spatial | 03-10.2016 | Global except SW quadrant | 107 736 |
| Temporal | 01-02 and 11-12.2016 | SW quadrant | 12 420 |
| Spatio-temporal | 01-02 and 11-12.2016 | Global except SW quadrant | 30 659 |
| Global | 12.2008 | Global | 7 111 |

**Table B.1 : Name, time period, spatial extent and number of samples for each of the five described data splits.**

We then train an AE model using the training data from the first data split (*random*). Each test data split is then used to evaluate the trained model through the reconstruction errors divided by the reconstruction error mean of the *random* split (noted as reconstruction error ratio; Fig. B.1). Spatial distribution of the mean reconstruction errors is shown in Figure B.2. We detail in Table B.2 the average channel reconstruction error for each of the splits.

We first notice that the reconstruction power of the model is consistent regardless of the test split considered with mean reconstruction error ratios ranging from 0.63 to 1.0, dividing the split's reconstruction error by the random data split mean reconstruction error. Ratios around 1 or below indicate that the model's performance is not inflated when considering a random data split, highlighting that the model did not only learn from possible spatial and/or temporal correlations between samples present in the training set. The distribution of the error is also very similar throughout the test splits with most of the samples located below an error ratio of 0.5. However, one of the main aspects regarding the performance of the model across test splits is the presence of a heavy tail in the distribution showcasing that for some samples the reconstruction error can be greater than 3 times the mean error. Looking at the spatial patterns of the reconstruction error, we note that overall the error comes from the COT and CWP predictions, the average reconstruction errors across test sets being 0.15, 0.32 and 0.25 for CTH, COT and CWP respectively (Table B.2). For the CTH, the error is concentrated in the zones with frequent convection around the equator and could be explained by local convection cells exhibiting a larger spread in CTH values. Another source of error could be that higher CTH values are also less represented in the training data. On the contrary, the error for COT and CWP is prevailing in high-latitude regions. Overall, the performance skill of the AE model seems to hold through the different test data splits. One could argue that the training dataset already retains enough variability in the data which could explain why the model still performs well regardless of the test set split. However, this consistent skill also shows that the performance reported in appendix C on the test set can be trusted to hold for other datasets and supports the data generation process to train the AE (cf. section 2.4).

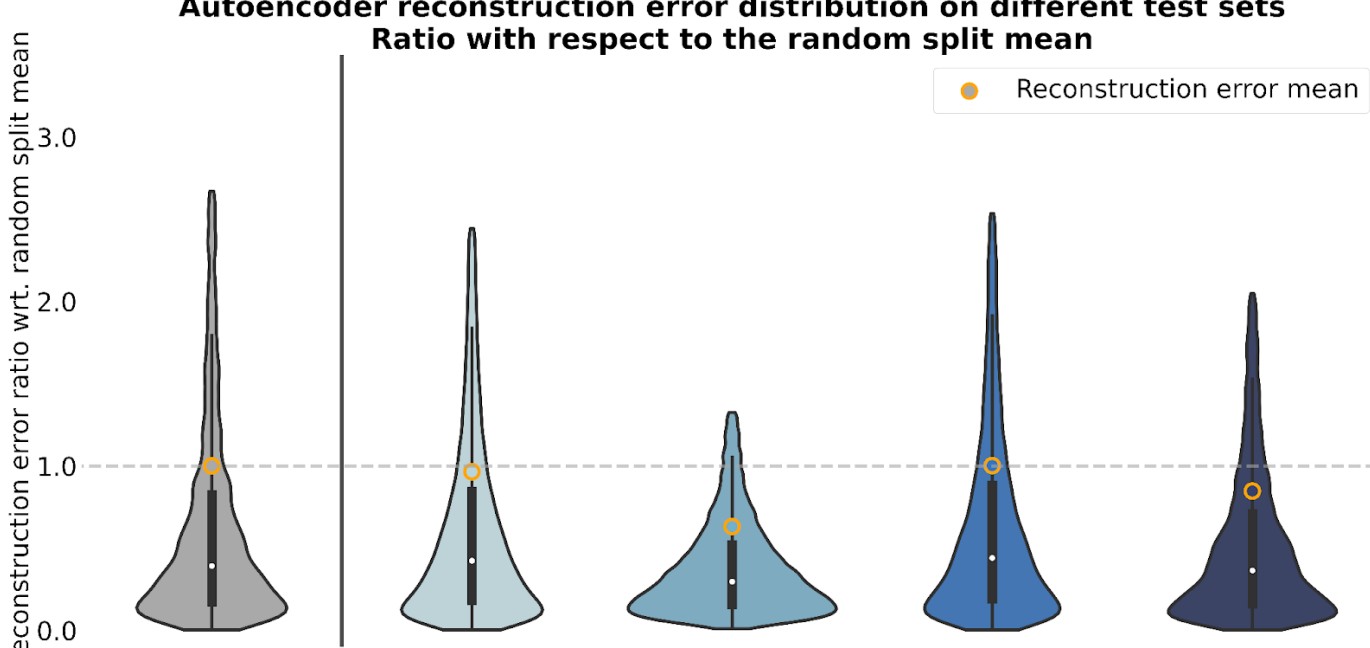

**Figure B.1:** Reconstruction error ratios of an AE on different test datasets. The quartiles are indicated with the barplot inside each violin plot while the mean is indicated with an orange circle. Extreme values were removed before plotting. Each sample's reconstruction error is divided by the mean reconstruction error of the random data split and defines the reconstruction error ratio presented here.

| Data split | Channel | | | Average |
|---|---|---|---|---|
| | **CTH** | **COT** | **CWP** | |
| Random | 0.117 | 0.369 | 0.333 | 0.273 |
| Spatial | 0.171 | 0.344 | 0.276 | 0.263 |
| Temporal | 0.114 | 0.253 | 0.150 | 0.172 |
| Spatio-temporal | 0.202 | 0.332 | 0.286 | 0.274 |
| Global | 0.154 | 0.318 | 0.221 | 0.231 |
| Average | 0.152 | 0.323 | 0.253 | 0.243 |

**Table B.2 :** Average channel reconstruction error for each of the five described data splits.

**Channel reconstruction error mean**

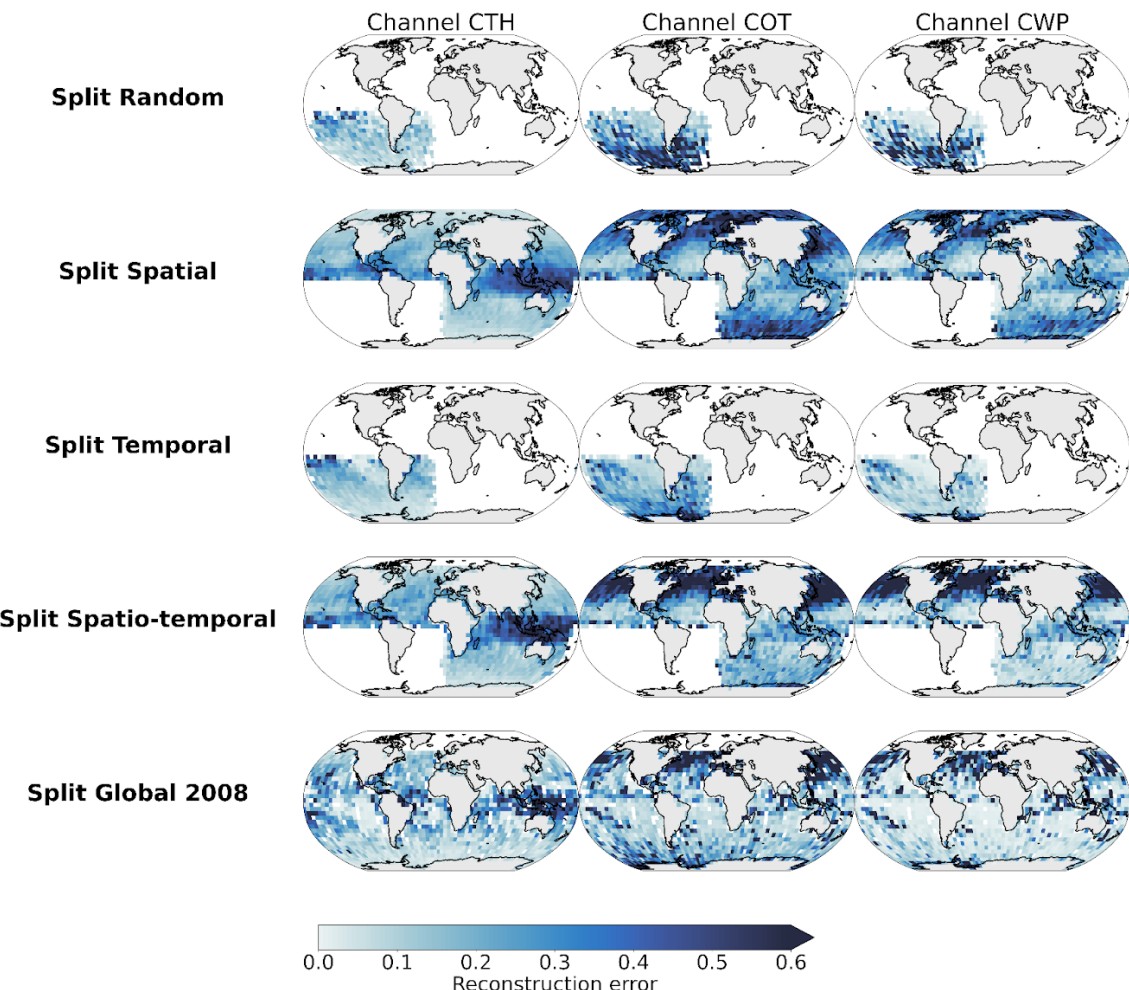

**Figure B.2: Distribution of mean channel reconstruction errors aggregated on a 5 ° grid.**

## Appendix C: Autoencoder architecture

| Layer | Hyperparameters | Output shape |
|---|---|---|
| **Input** | | (None, 3, 128, 128) |
| **Encoder** | | |
| Conv2d | (kernel = 3, stride = 2) | (None, 3, 64, 64) |
| ConvBlock x 5 | Conv2d (kernel = 3, stride = 1)<br>LeakyReLU<br>Conv2d (kernel = 3, stride = 1)<br>LeakyReLU<br>Conv2d (kernel = 3, stride = 1)<br>BatchNorm2d<br>LeakyReLU<br>MaxPool2d (kernel = 2, stride = 2) | (None, 256, 2, 2) |
| Flatten + Linear | | (None, 256) |
| **Decoder** | | |
| Linear + Unflatten | | (None, 256, 2, 2) |
| ConvTranspose2d | (kernel = 2, stride = 2) | (None, 256, 4, 4) |
| ConvTransposeBlock x 5 | Conv2d (kernel = 3, stride = 1)<br>LeakyReLU<br>Conv2d (kernel = 3, stride = 1)<br>LeakyReLU<br>Conv2d (kernel = 3, stride = 1)<br>BatchNorm2d<br>LeakyReLU<br>ConvTranspose2d (kernel = 2, stride = 2) | (None, 3, 128, 128) |

**Table C.1 : Autoencoder model specifications.**

| Hyperparameter | Value |
|---|---|
| Batch size | 64 |
| Epochs | 80 |
| Optimizer | Stochastic Gradient Descent (SGD), momentum = 0.9, learning rate = 0.0001 |
| Metric | MSE |
| Early stopping | patience = 20 |

**Table C.2 : Autoencoder model training specifications.**

**Appendix D: Ordinal regression**

We define our labels $y$ which can take values in $K = 9$ classes from $\{50\,\text{m}, 100\,\text{m}, \ldots, 2500\,\text{m}\}$. We introduce $K - 1$ thresholds $\alpha_y$ to define the separation of our $K$ classes which actually correspond here to the classes too. For each labelled sample $(s, y)$ the output of our model is $z = z(s)$. The correct interval for this sample is then $(\alpha_{y-1}, \alpha_y)$. During the fitting process, the goal is to find the set of parameters of our model $z$ and the corresponding thresholds $\alpha$ which minimises a certain cost function. We consider a generic nonnegative penalisation function $f(\cdot)$ (eg. hinge loss, squared error loss, Huber loss). There are then different ways to represent threshold violations and thus to penalise the predictor. While immediate-threshold setup only considers the thresholds of the correct interval, all-threshold setup takes into account all the threshold violations. In the case of an immediate-threshold setup the loss function would look like:

$$\mathcal{L}(z, y) = f(z - \alpha_{y-1}) + f(\alpha_y - z) \qquad (D.1)$$

Here we can see that the loss is not aware of how many thresholds are actually violated. In the case of an all-threshold setup the loss function is a sum of violations across all thresholds:

$$\mathcal{L}(z, y) = \sum_{i=1}^{K-1} f(t(i, y)(\alpha_i - z)) \qquad (D.2)$$

where $t(i, y) = -1$ if $i < y$ or $+1$ if $i \geq y$. Thus predictions are encouraged to violate the least amount of thresholds. We give in Figure D.1 an example of what the loss function would look like in the case of $K = 6$ labels and using a hinge penalisation.

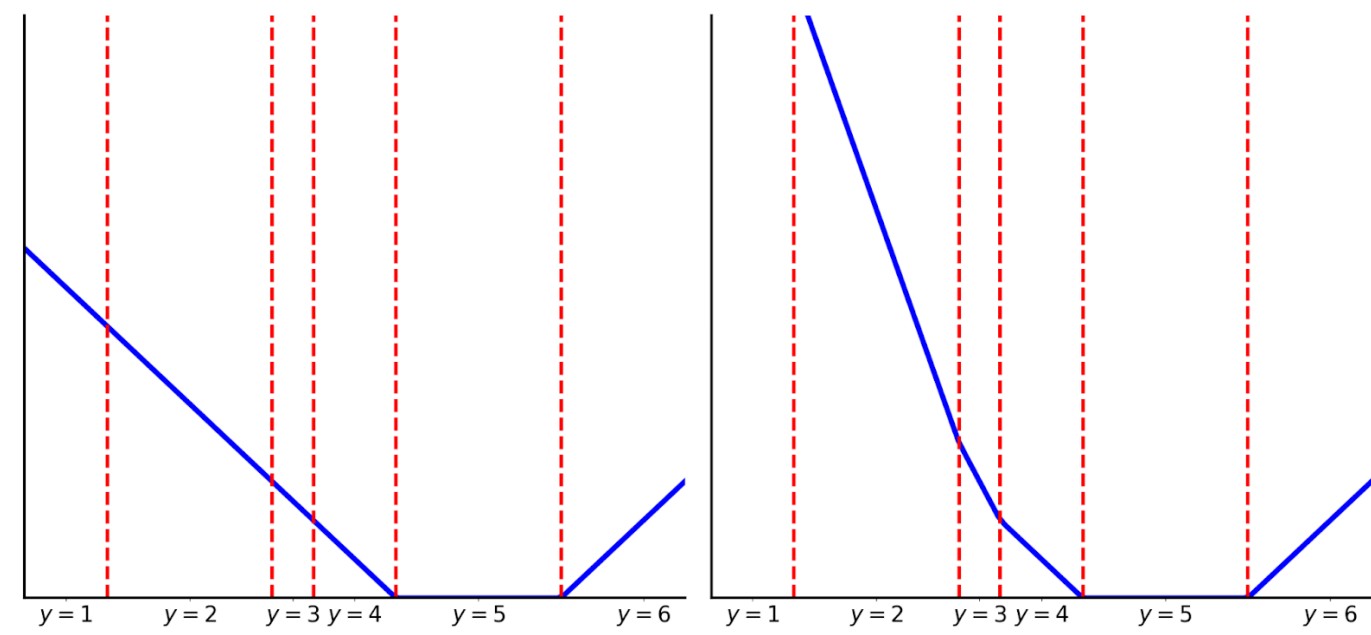

**Figure D.1: Threshold-based setups loss function representation for a hinge penalisation, K=6 labels and target label y=5. (left) Immediate-threshold and (right) All-threshold setup loss function. (figure adapted from Rennie et al. (2005))**

**Appendix E: Cloud base height retrieval method assuming adiabatic cloud**

Algorithm adapted from Goren et al. (2018). We use the retrieved CTH, CTT, CTP and CWP from MODIS MYD06 (Platnick et al., 2017).

---

**Algorithm:** Cloud base height retrieval

---

**Data:** CTH, CTT, CTP, LWP, look-up tables
**Result:** CBH

**if** CTT < 263.13 **then**
    **return** NaN
T ← CTT - 273.13
LWP obs ← LWP
LWP adi ← 0.
$\delta z$ ← 0.
Set corresponding cloud top indexes for temperature $T_{ind}$ and pressure $p_{ind}$ look-up tables.

Read-in the water mixing ratio w at the corresponding indexes.
**if** w out of look-up table **then**
    **return** NaN
**while** LWP adi < LWP obs **then**
    $\rho_{tmp}$ ← density look-up table with $T_{ind}$ and $p_{ind}$
    $\delta_{tmp}$ ← layer depth look-up table with $T_{ind}$ and $p_{ind}$
    $\delta z$ ← $\delta z + \delta_{tmp}$
    $w_{tmp}$ ← mixing ratio look-up table with $T_{ind}$ and $p_{ind}$
    LWP adi ← LWP adi + $(w_{tmp} - w) \times \delta z_{tmp} \times \rho_{tmp}$
    Adjust temperature T given the saturated lapse rate using look-up table with $T_{ind}$ and $p_{ind}$
    Update indexes $T_{ind}$ and $p_{ind}$
**return** CTH - $\delta z$

---

**Table E.1: Pseudo code for cloud base height retrieval algorithm assuming adiabatic cloud, adapted from Goren et al. (2018).**

**Code availability**

The code used for the method and producing the plots is available on Zenodo (Lenhardt et al., 2024).

**Data availability**

The global dataset of the cloud base height predictions for the year 2016 is available on Zenodo (Lenhardt et al., 2024). The dataset is available as a csv file with corresponding coordinates, MODIS granule, time of retrieval and predicted cloud base height or in a netCDF file as daily aggregates on a regular grid with a resolution of 1 ° or 5 °. The meteorological observations from the UK MetOffice (Met Office, 2006) are available through the CEDA archive at https://catalogue.ceda.ac.uk/uuid/77910bcec71c820d4c92f40d3ed3f249. The files from the CUMULO dataset (Zantedeschi et al., 2019) are available at https://www.dropbox.com/sh/i3s9q2v2jjyk2it/AACxXnXfMF5wuIqLXqH4NJOra?dl=0.

**Author contribution**

JL, JQ and DS designed the study. JL wrote the code. JL conducted the analysis and JL, JQ, DS interpreted the results. JL prepared the manuscript, JQ and DS reviewed the manuscript and provided comments.

**Competing interests**

The authors declare that they have no conflict of interest.

**Acknowledgements**

This work was supported by the European Union's Horizon 2020 research and innovation programme under Marie Skłodowska-Curie grant agreement No. 860100 (iMIRACLI). We thank the Leipzig University Scientific Computing cluster for computing and data hosting. We further thank Tom Goren for providing access to code snippets from Goren et al. (2018) and thank Olivia Linke for helping review the manuscript. We acknowledge the contributors of the CUMULO dataset (Zantedeschi et al., 2019) for providing access to the data files hosted at https://www.dropbox.com/sh/i3s9q2v2jjyk2it/AACxXnXfMF5wuIqLXqH4NJOra?dl=0. Additionally, we acknowledge the MODIS L2 Cloud product data set from the Level-1 and Atmosphere Archive and Distribution System (LAADS) Distributed Active Archive Center (DAAC), located in the Goddard Space Flight Center in Greenbelt, Maryland (https://ladsweb.modaps.eosdis.nasa.gov/archive/allData/61/MYD06_L2/). We would like to thank two anonymous reviewers for their constructive and detailed comments.

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
