# Peer review of "EGUSPHERE-2024-327 | Research article"

_EGUsphere, 2024_

## Author Comment (AC1)

**Response to Referee #2 on the manuscript "Marine cloud base height retrieval from MODIS cloud properties using machine learning"**

Julien Lenhardt, Johannes Quaas, and Dino Sejdinovic
31.05.2024

Dear anonymous referee,
We would like to thank you for the insightful comments and the constructive discussion. Please find our response below, in which the review comments are in bold and followed by our response.
In the revised manuscript, we include further details on the AE training and evaluation, gathering this information in a more centralised manner to hopefully better the understanding of the manuscript. Additionally, a more thorough presentation and evaluation of the ordinal regression model is incorporated to address the major comments. Overall, the comments regarding the coherence between sections and sentences were addressed, but the changes between the original and revised manuscripts are not included here as they are too substantial.

Best regards,

Julien Lenhardt on behalf of the authors

**Summary**

**This is a useful and straightforward paper that develops a new algorithm for estimating marine cloud base height from MODIS data, employing a machine learning technique. Evaluations against surface ceilometer observations and CALIPSO data demonstrate its superior performance over previous methods for cloud-base height retrieval. Furthermore, the resulting cloud-base height products are made publicly available on Zenodo, facilitating their utilization by other researchers within the community.**
Many thanks for this supportive summary of our study.

**General comments**

**While the methodology and results are robust and convincing, the paper's presentation suffers from several shortcomings. There is a lack of coherence between sentences and paragraphs, making it challenging for readers to follow the logical flow. Additionally, the frequent use of phrases like "It is to be noted that" disrupts the clarity of the text. Grammar errors, such as the phrase "allow to properly quantify" in Line 363, further detract from the overall quality of the paper.**
**Therefore, I recommend that the authors undertake a comprehensive revision of the language to improve coherence, eliminate ambiguous phrasing, and rectify grammar errors. This revision will enhance the paper's suitability for publication in ACP.**
We thank the reviewer for the evaluation of the manuscript. We hope that through the modifications made in the revised manuscript we addressed the language quality and improved the overall readability and coherence of the manuscript.

---

## Author Comment (AC2)

**Response to Referee #1 on the manuscript "Marine cloud base height retrieval from MODIS cloud properties using machine learning"**

Julien Lenhardt, Johannes Quaas, and Dino Sejdinovic
31.05.2024

Dear anonymous referee,
We would like to thank you for the insightful comments and the constructive discussion. Please find our response below, in which the review comments are in bold and followed by our response.
In the revised manuscript, we include further details on the AE training and evaluation, gathering this information in a more centralised manner to hopefully improve the understanding of the manuscript. Additionally, a more thorough presentation and evaluation of the ordinal regression model is incorporated to address the major comments. Overall, the comments regarding the coherence between sections and sentences were addressed, and for convenience, the edits can be viewed in the revised version of the manuscript which includes tracked changes.

Best regards,

Julien Lenhardt on behalf of the authors

**Summary**

**This manuscript presents a novel cloud base height retrieval algorithm based on Aqua MODIS level 2 cloud properties. Cloud base and cloud vertical extent are essential to determine the Earth's energy budget and reducing uncertainties would help to constrain studies of cloud regime dependent processes and to evaluate how clouds are represented in models. Therefore, the presented contribution is very welcome and in general fits the scope of AMT very well. Moreover, the use of machine learning poses a new perspective on satellite-based CBH retrievals which could help to overcome the challenge of determining subadiabaticity of clouds.**
Many thanks for an excellent and encouraging summary of the study.

**General comments**

**The presented approach utilizes a CNN-based autoencoder to reduce the dimensionality of input fields of size 128 km x 128 km which feature cloud top height, cloud optical thickness and cloud water path. In a second step an ordinal regression is performed on the latent feature vector produced by the encoder. Therefore, marine CBH observations from surface-based ceilometers which are available in discrete bins, are utilized as ordinal reference data. The study is limited to the ocean only to exclude potential complexity related to topography over land and the authors limit the retrieval to scenes with at least 30% cloud cover. In principle, I agree with the author's reasoning that spatial patterns of CTH, COT and CWP could be exploited to retrieve CBH. Overall, this could be an approach that further reduces uncertainties for satellite-based CBH retrievals while at the same time providing an interesting example of how to utilize machine learning for performance gains in satellite remote sensing of clouds. However, while I appreciate the compact form of the manuscript, I would like to encourage the authors to address the following major and minor concerns before considering its publication.**
We thank the reviewer for the very thorough and informative comments on the manuscript. We provide a detailed explanation of how we have addressed your comments individually and made corresponding adjustments to the manuscript.

**Comments**

**Major points:**

**I) Autoencoder setup:**

**The autoencoder (AE) role is to reduce the dimensionality of the input fields. The latent feature vector created thereby is then used as input for an ordinal regression onto ground-based reference CBH retrievals which are available in discrete height bins. While it seems promising to exploit the information inherent in the spatial distribution of CTH, COT, CWP using CNNs, I have the following concerns regarding the AE:**

**1. The information how the AE is trained and evaluated are distributed across different parts of the manuscript (i.e. Section 2.4, Appendix B, Appendix C) which makes it difficult to follow the data processing steps clearly. According to Section 2.4, the AE is trained on data from the year 2008. It is not clear which data is used for validation (e.g. in Fig. C.1). Furthermore, the performance on another test data set is not provided specifically for the trained AE which is ultimately used for the CBH retrieval model. This should be included in Section 2 to show the capabilities of the utilized AE. Maybe a figure showing the original and reconstructed CTH, COT, and CWP for one exemplary tile could be helpful to illustrate the process together with panels similar to Fig. B.2 but for training, validation and test data for the actual AE that is used later on.**

We thank the reviewer for the relevant comment. We adapted the structure of the paragraphs related to the AE training and testing. Indeed, the performance on the test set is missing and has been appropriately added alongside plots of reconstructions and channel errors as suggested.

**2. The selected architecture with the 5 convolutional blocks seems arbitrary. As this study proposes a novel retrieval algorithm, describing the algorithm development (together with validation) should be considered most relevant. Therefore, it is necessary to elucidate decisions, such as architecture, activation functions, loss function, etc., in more detail. The following issues should be addressed:**

**- Why was this particular architecture chosen (i.e. the 5 Conv. blocks with 3 conv. layers and a max pooling layer each, Table C.1)? Were other combinations tested?**

**- Why are LeakyReLU and MSE chosen as activation and loss functions? Have others been tested? So far, the choices appear arbitrary.**

**- The validation data should be used to tune the architecture (i.e. number and type of layers, loss function, activation functions, learning rate, etc.) with the goal of minimizing some performance measure (e.g. MSE). The test data should shed light if the final setup generalizes to independent data.**

Some details were indeed omitted and are indeed required in this section. We thank the reviewer for picking that up and we now describe in more details the model in the corresponding section. To answer specifically the different points in the comment:

- The architecture is a direct adaptation of the very common VGG net from Simonyan and Zisserman (2015) and the U-Net architecture from Ronneberger et al. (2015) where the block structure of the model is used. The size of the convolution filters is additionally justified in these papers.
- LeakyReLU has been shown to improve model performance compared to ReLU (Nair and Hinton, 2010) which is used in the U-Net architecture for example.
- The MSE choice for the loss function was how AE were introduced (Kramer et al., 1991). Typically another choice would otherwise be the Binary Cross Entropy (BCE) loss but since our data in this case is represented by floating point values and not RGB pixels (which are constrained to the interval [0, 1]) we decided to stay with the MSE loss.
- No fine-tuning of the architecture was made here since the transferability of such models and architectures has been proven over time in many applications (Reichstein et al., 2019).

- We finally added the performance on the test set and on a test set from a different year (2016)

**3. The advantage/benefit of the AE in leveraging the spatial information and its sensitivity to the input tile size has not been shown so far. The AE reduces an array of 3x128x128 to a vector of 256 values. The size was chosen "to give enough spatial information to the AE" (l. 166). But what is enough spatial information and how sensitive is the reconstruction error and more importantly the performance of the ultimate CBH retrieval to this size? The authors argue that similar spatial scales were used in other studies. However, for instance, Mülmenstädt et al. (2018), and Lu et al. (2021) extrapolated the signal from thinner clouds into a larger domain and Böhm et al. (2019) inferred the CBH from the distribution of CTH within a cloud field. While the assumption of a homogeneous CBH within a certain area is the same, the objective to utilize a certain spatial scale differs from this study. Here the question is, how much spatial information is needed to reproduce the input cloud properties. How about using CTH, COT, CWP fields from 9x9 tiles around the reference CBH to perform the ordinal regression directly without use of an AE? That would have a similar dimensionality (3x9x9=243). Thereby, a potential benefit of the AE could be assessed making a stronger argument for the choice. Together with some statistically based arguments on the tile size, the method development will be more stringent.**

The reviewer is right and we take this comment into account in two separate ways:

- As an ablation study, we now include a baseline method using a 9x9 tile size with no use of the AE to compare to in the corresponding section under "Spatial context". We present below the two confusion matrices obtained on the validation set for this new baseline method (right) and the initial presented method (left). The conclusion seems quite clear that the spatial information condensed and encoded by the AE helps retrieving with higher quality the CBH. Considering the performance metrics, the baseline 9x9 tile method yields a MA-MAE of 803 m and a MA-RMSE of 1186 m compared to 661 m and 796 m respectively for the presented method with spatial context. The ascertainment is the same when looking at the OC, UOC and AUOC where the baseline method is systematically outperformed, displaying a higher index/metric (0.05 for each of the three metrics).

[Figure]

- We add some arguments and discussion regarding the influence of the tile size in the following section under "Tile size". Overall the larger tile size of 128 provides better performance across CBHs (MAE and RMSE notably) and is thus kept in the rest of the study.

**4. To exploit the input features most efficiently in problems with sparse reference labels, I would argue, that a self-supervised pretraining with subsequent finetuning could result in better overall performance. That way, the trained AE could actually be geared towards CBH retrieval and not just towards reconstructing the input fields. Fabel et al. (2022, https://doi.org/10.5194/amt-15-797-2022) show an example for cloud layer classification (low/medium/high) from all-sky cameras. While they do not consider the ordinal nature of their**

**classes, there are also some works that find targeted neural network solutions for ordinal target data (e.g. Lazaro and Figueiras-Vidal, 2023, https://doi.org/10.1016/j.patcog.2023.109303). I do not expect the authors to change to such methods, as long as it can be shown that the use of an AE actually adds value and that there is some sensitivity towards the tile size which would imply that the AE actually leverages the spatial structure of the input fields leading to an improved CBH retrieval.**

Thank you for these insightful suggestions, we agree that these ideas are an important topic for further study. We explored using the CORAL (COnsistent RAnk Logits; Cao et al., 2020, https://doi.org/10.1016/j.patrec.2020.11.008) and CORN (Conditional Ordinal Regression for Neural networks; Shi et al., 2021, https://arxiv.org/abs/2111.08851) frameworks during the experiments of the model but decided against since we had trouble making the training converge. However, we only tried these networks by adapting the last layers of the AE model and fine-tuning them. Maybe it would have rather been required to train the whole network weights again with the adapted last layers. We now discuss these alternative approaches to working with ordinal data using neural networks in the revised manuscript.

**II) Ordinal regression:**
 **1. What parameters are fitted by minimizing the loss is missing. Furthermore, it is unclear which data set (time period, region) is used to train the OR model. Next to the performance on the test data set (Table 2), the performance should also be shown for the training data to see if there is overfitting.**

Thank you for pointing out the missing details which will improve reproducibility of our results. The manuscript was adjusted accordingly.
 **2. The authors investigate the ability of the AE to generalize on unseen spatial and temporally independent data. While this is interesting, it is further and even more necessary to show the generalizability also for the final CBH retrieval model which includes both encoder and OR. The authors argue that testing the spatial generalizability is challenging due to sparse distribution of collocated retrievals and that "Limiting the training dataset to a selected area would greatly hinder the representativeness notably because the different labels display diverse spatial patterns." (l. 283-284). Would it be possible to train on the northern hemisphere and test on the southern hemisphere and evaluate the performance for each CBH bin seperately? That should ensure that the training data includes sufficient samples from all CBH bins and that a potential different CBH distribution in the test data does not yield an apparent performance difference. If more data would be required, an option would be to obtain MYD06 data directly from NASA's Level-1 and Atmosphere Archive & Distribution System Distributed Active Archive Center (https://ladsweb.modaps.eosdis.nasa.gov/archive/allData/) which goes beyond the 2 years included in CUMULO.**

The reviewer is right that investigating the spatial generalisation not only for the AE but also for the CBH prediction task is necessary. To this extent we followed the recommendation and split the labels into Northern and Southern hemispheres subsets, subsequently training a model on one hemisphere and testing it on the other one. We report the performance metrics in the plots below. This experiment has been added to the manuscript in further detail. However, due to data storage limitations it is rather challenging to perform the colocation process for a wider data period. We deemed relatively satisfactory the amount of labels gathered for this two-year period, even though incorporating further labels will likely yield better performance for the developed retrieval method.

[Figure]

3. To obtain a reference, the authors adapt CBH retrieval methods based on an adiabatic cloud profile (adapted from Goren et al., 2018) and based on a statistical relationship between cloud geometrical thickness on one hand and CTH and CWP on the other (adapted from Noh et al., 2017). While they could show that their retrieval can achieve lower errors (Table 2), it remains unclear whether the method outperforms a trivial approach. While the overall bias is close to zero, the method underestimates higher CBHs and overestimates lower CBHs (Fig. 3) which indicates that the algorithm favors CBH values closer to the mean of the reference CBH distribution. Therefore, two things should be investigated to really show the benefit of the proposed method:

- I would argue that the utilized error metrics are not sufficient to investigate the skill of an algorithm. A model with a high MAE could still have skill if the ordering of the retrieved CBH resembles the ordering of the reference CBHs. While I agree that standard correlation indices are not appropriate for ordinal data, there are some metrics proposed, such as the ordinal classification index (Cardoso and Sousa, 2011, https://doi.org/10.1142/S0218001411009093) which was updated by Silva et al. (2018, https://doi.org/10.1109/IJCNN.2018.8489327) that could help assess the skill and compare different methods.

We thank the reviewer for this useful remark. We implemented both metrics described in Cardoso and Sousa (2011) and Silva et al. (2018) to help better evaluate the model. We notice one caveat though in the developed metrics as they treat each class as equally distant (the term $|r - c|$ representing the distance to the diagonal in the confusion matrix). While this may be the case in ordinal classification, using such metrics in the context of ordinal regression could lead to misrepresentations. For instance, in our study the difference between a CBH of 200 m being misclassified as 50 m (difference of 2 classes or 150 m) error should not yield a similar penalization as a CBH of 1500 m being misclassified as 600 m (difference of 2 classes too but 900 m error). But a purely ordinal classification index will drop all information on the scale of the response (1500 misclassified as 600 treated the same as 200 misclassified as 50, since only the order matters) which might be not entirely appropriate for this problem. We adapted the metric as of consequence, representing the data as if the CBH classes were regularly spaced from 50 m up to 2500 m to mimic the span between CBH classes.

- How does the algorithm compare against a trivial (always choose the majority bin) and semi-trivial retrieval (always choose the bin for which the MAE or RMSE are minimized when this bin is chosen for all samples)?

It is indeed always useful to integrate a trivial method to compare to and we thank the reviewer for pointing that out. In the case of our study, the majority bin is the 600 m one and is additionally the one for which the MAE is minimised when selecting it for all the samples (over the training set). We included the comparison of the two (in the end only one because of the same bin being chosen) trivial methods along the 9x9 tile suggestion in the same section discussing the spatial context leveraged, or not, by the use of the AE.

**III) Global distribution:**

**The developed CBH retrieval algorithm is applied to assess the global distribution of CBH for the whole year of 2016. However, for this application the authors should better use the previously trained and evaluated model for error statistics are known instead of retraining the model with different data.**

We thank the reviewer for this important point and have updated the manuscript accordingly. We additionally re-run the processing of the global dataset with the previously trained model instead of the retrained model.

**Minor points:**

**• In the introduction (specifically I. 61ff), the authors should state more clearly what their goal is (i.e. CBH retrieval with reduced uncertainty) instead of saying the "developed ML model aims to draw on the spatial information [...]" (I.62). They should state why they expect their ML approach to be superior, e.g. through a hypothesis that the spatial pattern of a cloud field holds information on the CBH and the potentially non-linear relationship to satellite observations can be exploited through a convolutional neural network.**

We agree that the emphasis should be put on the targeted reduced uncertainty of the retrieval along with the potential benefits of the ML model. This passage has been adapted in the revised manuscript.

**• the structure of Section 2 (Data and methods) should be improved. I suggest:**
**  - Briefly describe the overarching idea of the approach in the introduction:**
**    - building a CBH retrieval using level 2 satellite data (CTH, COT, CWP)**
**    - self supervised training of a CNN applying an autoencoder**
**    - subsequent ordinal regression using ground-based marine CBH retrievals**
**  - Take out the first 2 paragraphs in Section 2 (the text before Section 2.1)**
**   - Instead, put all details on the ground-based and satellite data as well as the methods in the corresponding section (Section 2.1 to 2.5)**
**   - Information from Appendix C should be merged into the main text (i.e. into Section 2.4). It is partly repeating points already mentioned in Section 2.4.**

We thank the reviewer for the suggested structure. We agree that the information was probably not well partitioned between the different sections and appendices. To this extent, we updated the contents of the introduction, section 2 (Data and methods), appendices B and C in the revised manuscript.

**• It should be clarified what the reconstruction error is actually. In I. 214 it is mentioned that an l2-norm would be a common choice. It should be clarified that this is also applied here. Next to Table C.2 which states that a MSE is used as loss function, terms such as "reconstruction error ratio" (Fig. B.1), "reconstruction relative error" (Table B.2) and "reconstruction error" (Fig. B.2) are used. Providing an equation to define the reconstruction error as a variable would avoid confusion.**

The reviewer is right and a clear mention of the reconstruction error, which is indeed the MSE, has been added in the AE section to avoid any confusion through the manuscript.

**• "the binning process can lead to an underestimation of the actual CBH" (I. 113) - Has this been shown? Would not an overestimation also be possible?**

The base of the binning process can in its nature lead to an underestimation as the reported bin label will always be lower or equal to the actual measurement. For example, a ceilometer observation of 2499 m would be binned into the 2000 m category. On average we can only expect the actual retrieval made available in the dataset to be an underestimation of the real measured quantity. Since the pre-binning measured value is not available, it remains impossible in this case to quantify the bias in the estimated CBH retrievals. For more clarity, the sentence was extended to include this explanation:

"Since the actual measured CBH values are not available in the dataset, it is impossible here to directly quantify the possible bias stemming from this binning process. In general here, we can suspect that the available CBH retrievals represent an accurate or underestimated assessment of the effective CBH, as for example a ceilometer measuring a CBH of 2490 m will be reported in the 2000 m bin in the available dataset.".

**• Use of the term "swath": The term "swath" refers to the width of the instruments view of the Earth surface (e.g. MODIS swath is 2330 km). I think the authors should exchange their usage of "swath" by "granule". A MODIS granule is the information stored in one MODIS file and it covers 2330 km x 2000 km.**
Thank you for this terminology clarification, we replaced in consequence the term "swath" in the manuscript when necessary.

**• Satellite data description should include more details on the MODIS retrieval of CTH, COT, CWP. For instance, it should be mentioned that they require additional input such as temperature, water vapor and ozone profiles from NCEP GDAS (e.g. Platnick et al., 2003, https://doi.org/10.1109/TGRS.2002.808301; or Baum et al., 2012, https://doi.org/10.1175/JAMC-D-11-0203.1) as this has implications on potential uncertainties in particular in remote marine regions with sparse observations available for assimilation.**
Thank you for this suggestion, we have added further details as requested in the corresponding section.
**The study focuses on low clouds for which the CO2 slicing technique fails. It should be outlined, that the CTH retrieval is then based on the 11µm brightness temperature combined with simulated BTs based on vertical profiles from GDAS using surface temperature together with a monthly averaged lapse rate (Baum et al., 2012, https://doi.org/10.1175/JAMC-D-11-0203.1). It would be helpful, if the authors could comment briefly on potential impacts on seasonally and regionally changing biases that might relate to such derived MODIS CTHs.**
We thank the reviewer for this relevant insight regarding the satellite retrieval pipeline. We included some comments in the revised manuscript.

**• MODIS tile around surface observation: The description in Section 2.3 implies that the 128 km x 128 km MODIS tile is selected so that the surface observation is located at the center of the tile but it is not explicitly mentioned. Or could the surface observation also be closer towards the edge of the tile? Maybe that could be clarified.**
The surface observation is indeed located at the centre of the extracted tile and includes the granule data for the whole 128 km x 128 km tile around. It is now clarified in the manuscript.

**• For the tile size, the authors "compromise between considering all the relevant information while not discarding too many samples which might fall outside of the distance limit." (l. 168-169). Does that mean, the 128x128 is big enough to include relevant spatial information but small enough, so that not too many samples have to be discarded because the tile size would exceed the region covered by the MODIS granule? Maybe rephrase the sentence, to be clear.**
We rephrased the sentence to clarify the statement.

**• Cloud cover filter: The authors filter for MODIS tiles with cloud cover >= 30% and mention that a lowering of this filter does not improve the performance. Would one not aim for a filter as low as possible to be able to apply the developed algorithm to as many scenes as possible? In other words, it would be more interesting to see if algorithm performance would decrease if the threshold is lowered. Ideally, the authors would find a minimum value for which the performance still holds but would allow to include more scenes.**

We thank the reviewer for this relevant comment. We added some more concrete evaluation of the cloud cover threshold in the section about the ordinal regression model. Overall, we choose to stay with the 30% cloud cover threshold as the lowering does not show fully convincing improvements on performance and only a marginal increase in the amount of samples. In our opinion, the best way to improve further the method with more samples would be to widen the colocation process. Below are the confusion matrices mentioned in the revised manuscript.

[Figure]

**The authors mention that the aim of the cloud cover filter is to avoid missing values. If that is the case, why not filter for scenes with a certain minimum portion of valid retrievals directly? Anyhow, it should be stated clearly how scenes with missing values in the CTH, COT, CWP are treated.**

Further clarification needs to be made regarding the missing values and cloud cover filters. We concordantly perform a filtering for the cloud cover using the cloud mask product (MYD35) and a filter for missing values on the cloud properties. Sometimes we experience missing values in one of the cloud properties (mainly COT and CWP when the retrieval failed entirely) and thus want to filter out these scenes. We clarified that now in the revised manuscript. The cloud cover is as described in the response to the previous comment.

**• To illustrate the functionality of the AE, it would be helpful to show an example of an original MODIS tile and the corresponding reconstruction. This could possibly be combined in a multipanel plot together with the training and test loss (Fig. C.1) and be placed in the subsection with the AE development (Section 2.4).**

A reconstruction plot was added in the manuscript as described in the response to the previous comment.

**• In Section 3.1 the AE-OR CBH retrieval is compared to methods developed by Goren et al. and Noh et al. The respective input data and how these methods are implemented here should be stated more clearly and be placed somewhere in Section 2. Furthermore, potential differences from the original implementations of those methods should be mentioned. For instance, Noh et al. applied their method to VIIRS.**

We agree and mention in more details the implementation of the two methods and their particulars in the section about CBH prediction (2.5).

**• Phrases like "fails at predicting with good accuracy" (l. 254) should be avoided. It is not clear what is "good" or "bad" accuracy. The question is, is one retrieval more suitable for a desired application than another.**

We thank the reviewer for the language clarification and have updated the manuscript accordingly.

**• The authors could also comment (maybe in the conclusion) if they would consider the AE-OR CBH retrieval to work on instruments other than AQUA MODIS.**

We added a comment regarding potential application to other satellite instruments in the conclusion.

**• Equations should be numbered (e.g. in Appendix D) so they can be referenced in the text**

Indeed, this was updated in the manuscript.

**• 2nd Equation in Appendix D: alpha_y -> alpha_i (c.f. Rennie et al. Equation 13)**
This typo has been updated, thank you!

**Additional specific/technical comments**

**l. 59 "Subsequent uncertainties" -> Subsequently, uncertainties**
We updated the manuscript following the suggestion.

**l. 59 "can then relate to uncertainties" -> propagate into uncertainties**
It indeed reads better with this suggestion, thank you!

**l. 62 "using an innovative machine learning (ML) model" -> using a machine learning (ML) model**
It has been modified in the updated manuscript.

**l. 64ff "As the CBH is typically derived from the surface": Consider rephrasing. It should be stated that more accurate CBH retrievals are obtained through ground-based remote sensing which are only available at isolated locations but can serve as reference data to develop satellite-based retrieval algorithms.**
This recommendation was adapted in the manuscript.

**"we focus on lower clouds in particular as the retrieval quality is generally higher": I guess, this refers to higher accuracy for ground-based CBH retrievals compared to satellite-based estimates but it is not clearly stated.**
Indeed, we clarified that.

**l. 107ff "At the beginning of meteorological [...]" - Due to its content, this sentence should already start the 2nd paragraph which describes the ground-based CBH retrieval.**
The reviewer is correct, this has been updated.

**l. 114-115 "[...] the surface-based observations specify quantities like temperature, humidity and wind speed [...]" - If the listed quantities are not used in the study, this sentence should be removed. Else it should be stated, what they are used for.**
This has been removed.

**l. 116-117 This paragraph turns back to geographic location of the ground-based CBH observations -> move to 1st paragraph of Section 2.1**
Moved the paragraph to the first part of the section.

**l. 130 "from the AQUA satellites" - from the AQUA satellite**
Indeed, thank you!

**l. 168-169 "compromise" - the two things the authors "compromise" between seem to pull in the same direction: make the scene larger -> consider more information + obtain more collocations -> that is not what is meant (the trade-off is the representativeness of the ground-based CBH retrieval) -> maybe rephrase**
This was accordingly rephrased in the revised manuscript.

**l. 173 "The extracted tile is then filtered" -> start new paragraph**
Adapted.

**l. 174-175 "The latter condition is primarily aimed at retrievals of poor quality" - maybe change to: "The latter condition is primarily aimed at avoiding retrievals of poor quality"**
The corresponding passage was adapted.

**l. 182-185 "future avenues of research could consider directly modelling unmatched datasets" - This is rather vague. Consider removing this paragraph.**
We reworked this passage as we think it is generally of relevance regarding the typical colocation processes done in climate research. Developing methods for which not all variables need to be known to still be able to produce a retrieval of quality seems a great way to still make use of the vast amount of observations available. It is shown in Lun Chau et al. (2021) in particular for downscaling applications how we can make use of mediating variables to achieve such a goal. It was however moved in the conclusion as an outlook on the colocation process and semi-supervised framework of the method.

**l. 213 "The main goal of the AE is to minimise the loss function" - The main goal is the dimensionality reduction. Maybe state "The main goal of the AE training [...]"**
Rightfully so, it has been updated.

**l. 215 "A common choice for the reconstruction metric is the ℓ 2- norm" - The authors should state that this is also what they chose to apply here.**
Updated accordingly.

**l. 252-253 "later on we refer to the overall conceived method [...] as OR + AE, interchangeably as OR or as the prediction model" - This could be simplified by giving the model one name and staying with it. How about ORABase?**
We thank the reviewer for the interesting suggestion as it could greatly improve understanding throughout the paper. We updated the method name in the revised manuscript where necessary.

**l. 311 "2B-CLDCLASS-LIDAR retrievals closer to the surface are not well captured" -> I think it is supposed to mean, the CBH is not well captured by the retrieval**
Formulation was adapted to clarify this.

**l. 317 "prediction model achieves similar performance as presented in Table 2" -> What does "similar" mean? The performance should be stated quantitatively.**
The quantitative performance metrics have been added.

**l. 335-336 "We then spatially aggregate the predictions over the year" -> It should be mentioned what the target pixel size is for the averaging. Furthermore, it is also a temporal averaging, so not just spatial aggregation.**
Indeed, a clarification should be mentioned here regarding the target resolution. To add to the temporal aggregation, we simply added the term "annual mean" in the sentence to clarify that. The two modifications have been included in the revised manuscript.

**l. 338 "more than 100 CBH retrievals over the year are displayed thus impacting mostly coastal and polar regions" -> displayed where? If you refer to Fig. 5, it seems that values are displayed for all ocean pixels, not just coastal and polar.**
The sentence could be written in a clearer way so the message is more easily understandable. What was meant here is that using such a threshold on the number of retrievals per grid cell for the whole year has mostly an impact for coastal and polar regions where less retrievals have been made throughout the year because of the ocean region filter. We decided to adapt the sentence to: "For more robust evaluation and statistics, only ocean grid cells with more than 100 CBH retrievals over

the year are displayed thus impacting mostly coastal and polar regions where filtering for ocean-only scenes or the original amount of satellite retrievals leads to a higher rate of displaying removal.".

**l. 363-364 "Using the spatially-resolved information of cloud fields with passive satellites allows to properly quantify lower cloud bases, more specifically avoiding the noisy retrievals of active satellites closer to the surface." -> I think what the authors are trying to say is: Using the herein described CNN-based autoencoder to process spatially-resolved CTH, COT, CWP results in more accurate CBH retrievals compared to the 2B-CLDCLASS-LIDAR product. I mean, using observations from active sensors could probably also be applied to train some retrieval algorithm which performs better than products currently available. But that is not part of this paper. Therefore, such phrasing needs to be revised. Also "avoiding the noisy retreivals" is not precise. It should be stated that certain performance measures are better for the developed retrieval method compared to other products which have been considered for this study.**
This sentence indeed needed rewriting to make the context and goal of the study clearer. We changed it including the previous recommendations to: "Using the spatially-resolved information of cloud fields of CTH, COT and CWP through the described CNN-AE results in more accurate CBH retrievals compared to the active retrievals of the 2B-CLDCLASS-LIDAR product, producing indeed better performance metrics compared to the other products and methods considered in this study."

**l. 364-365 "A CNN proves to be valuable to leverage spatial information" - This suggests that the convolutions play a key role. However, that has not really been shown. For instance, one could train a multi-layer perceptron to process pixel-based information to retrieve the CBH for each pixel and then calculate a field average or regress the px values onto the ground truth. Maybe that would produce similar results. All that has been shown was, using a CNN in an AE to reduce the dimensionality with a subsequent OR results in a better CBH retrieval compared to previous methods.**
The phrasing was indeed too broad as the performance was evaluated against other retrieval products but not a non-CNN based model using the same dataset from the study. We accordingly rewrote this passage to properly include the conclusions which actually fall in the scope of the presented manuscript.

**l. 465 "The correct interval for this this sample is then ($\alpha$ ¿ ¿ $y-1$ , $\alpha$ $y$ )¿" -> 2x "this" and formatting of the interval need to be revised**
Thanks for catching the typo and the formatting error, the manuscript has been revised, including the corrections.

**Fig. 5 MAD CBH shows a white (i.e. missing value) pixel in the South Atlantic (bottom panel), which is filled for the mean CBH (top panel)**
After re-running the global predictions with the trained model as per the related major comment, the null MAD pixel is not present anymore.

**Fig. D.1 Is it correct that the right panel shows the same loss for y=4 and y=5?**
Thank you for spotting this typo, the figure was corrected.

---

## Author Response (AR2)

**Response to Referee #1 on the manuscript "Marine cloud base height retrieval from MODIS cloud properties using machine learning"**

Julien Lenhardt, Johannes Quaas, and Dino Sejdinovic
16.07.2024

Dear anonymous referee,
We would like to thank you for reviewing the revised manuscript and providing additional comments. Please find our response below, in which the review comments are in bold and followed by our response.

Best regards,

Julien Lenhardt on behalf of the authors

**Minor/technical comments**

**It should be introduced why the reconstruction error, introduced as MSE (l. 211), does not carry a unit. If it is calculated as MSE between output and input field of the AE, it should carry the squared unit of the input variable. Or does it refer to the normalized input? This should be clarified.**
The error is indeed applied to the normalised inputs. We added at the end of line 214 in the revised manuscript:
"The MSE considered here between the inputs and outputs of the AE is unitless, as the inputs are standardised before processing to ensure each channel is on similar scales and a more stable model training."

**The line numbers refer to the tracked changes document.**

**l. 23-25: This implies that the test data set consists of both ceilometer and CALIOP CBH retrievals. I guess, that is not what is meant. Furthermore, the phrase "performs well on both datasets" is not helpful. A more quantitative statement is necessary.**
The word "evaluated" was included here to emphasis that the method is trained on the surface ceilometer measurements (line 21) but then evaluated using both surface measurements and CALIOP satellite retrievals. The following sentence at lines 23-25 was modified to:
"The statistical model performs similarly on both datasets, and notably on the test set of ceilometer cloud bases where it exhibits accurate predictions in particular for lower cloud bases and a narrow distribution of the absolute error, namely 379 m and 328 m for the mean absolute error and the standard deviation of the absolute error respectively."

**l. 71-73: "The objective of the developed method is primarily to produce CBH retrievals with reduced uncertainty, and additionally to extrapolate CBH retrievals from local surface observations to a wider spatial and temporal coverage" - Sounds like the ground-based observations are used as input and then together with satellite data extrapolated into space. However, they are only used to train the algorithm.**
Indeed, the sentence was reformulated to:
"The objective of the developed method is primarily to produce CBH retrievals with reduced uncertainty, and additionally to provide extended spatial and temporal coverage compared to surface observations."

**l. 129-130: One sentence alone should not constitute its own paragraph. Possibly connect it to the following paragraph.**

This sentence is actually the first one of the following paragraph and the corresponding pilcrow was already removed in the track changes manuscript (see last character of line 130).

**l. 131: Right before, it is stated that earlier reports are based on human observers. Now it is stated "The CBH is derived using a ceilometer". Whether the authors utilize only CBHs retrieved by ceilometers should be clearly stated.**

A clarification was added:

"In the surface observation dataset used in the study, the CBH is derived using a ceilometer, [...]"

**l. 136-141: From Fig. 2b the binning of the utilized ground-based CBH data set becomes clear. However, according to this text passage, the authors use the minimum of the bin range for their evaluation. Other than introducing a negative bias, it is unclear how this procedure affects the results.**

From choosing to use the bottom value of the bins as label, no real impact would be expected on the CBH predictions as they are considered as classes by the ordinal regression model. Naturally, some of the regression metrics would change (MAE, bias, RMSE). However, no clear comment could be made regarding for example the distribution in Figure 7 where the average might vary depending on the chosen labels.

**l. 148-149: Table 1 caption - references to Section 2.1 and 2.2 seem to be mixed up.**

Thank you for catching the typo.

**l. 187-189: "The level 2 product [...]" - sentence can be removed.**

This sentence is indeed a bit redundant with the previous paragraph and was thus removed.

**l. 190: "in particular" - these words can be removed.**

Removed.

**l. 207: "Appendix B" -> "Appendix C" (I suppose is meant here)**

Indeed, it was corrected.

**l. 265-278: The structure of this new paragraph should be improved. First, they say, data are only taken for the year 2008. Then they say, data are only taken for a single year to avoid correlation between training and test data. Then they say they obtain 500.000 samples that are split into training, validation and testing based on retrieval date. So are all these 500.000 samples from the year 2008? And then how are they split into train, validation and test sets? Adding to the confusion is the statement regarding another test set using data over land for the year 2016. Please, clearly state, which period and location are used for training, validation and test, respectively.**

The training, validation and testing datasets are created for the year 2008 and then split following their retrieval date in chunks of 70%, 20%, 10% "The overall built dataset consists of around 500 000 samples which are then splitted for training, validation and testing based on their retrieval date" (l271). To further test the generalisation of the model to unseen data, we build a similar dataset for the year 2016 but for which no train/val/test splitting is done "For further testing, Wwe additionally create a test dataset based solely on data from the year 2016 for further testing which includes tiles not only over ocean but also over land, indicating potential generalisation skill for unseen data including orography influence" (l272). The spatial distribution of the error is then reported for these overall 4 datasets "The spatially averaged reconstruction errors per cloud property channel are displayed in Figure 4 for each of the training, validation and testing datasets previously mentioned" (l275).

**l. 354: It should be added that the 9x9-tile-simple method still features the OR. However, instead of using the AE feature vector as input, it uses the 3 cloud properties within the 9x9 tile simply as a flattened vector as input. At least, I assume that is what is done.**

A OR model was indeed used with the 9x9 flattened tile data. This was clarified by adding in the sentence at line 354:

"[...] and an OR method relying on the flattened cloud properties of a 9x9 tile centred around the observation [...]".

**l. 357-361: The phrasing appears a bit complicated. The term "the baseline model" is not really defined. There appear 3 "baseline" models if you want (2 trivial, one 9x9 tile) so using this term is a bit confusing. And the error metrics for the two trivial methods should be identical since both always predict the 600m bin. Maybe it would be easier to just state the MA-MAE, and MA-RMSE for the developed AE-OR method and then in increasing order the errors for the other (9x9 method, trivial methods).**

The phrasing and label for each mentioned method was clarified in this section.

**l. 431-434: First, it is stated that the AE-OR method with immediate-threshold setup (IT) has similar (low) skill compared to the other retrieval. It is also stated, that the AE-OR method with all-threshold setup (AT) performs much better "on par with the other retrievals". If IT and AT methods differ, they cannot both be similar to the other retrieval method. Consider rephrasing this part.**

This part was rephrased with the following:

"We first note that the OR method with an immediate-threshold setup fails at predicting adequately the cloud scene base height compared to all the other retrieval products, producing large errors (double-fold in comparison to the all-threshold setup). On the other hand, ORABase performs well with satisfying error measures and uncertainty in the predictions, on par if not better than the two retrievals from Goren et al. (2018) and Noh et al. (2017)."

**l. 525 - "pacific" -> Pacific**

This was corrected accordingly.